# THE UNIVERSAL APPROXIMATION POWER OF FINITE-WIDTH DEEP ReLU NETWORKS

## ABSTRACT

We show that finite-width deep ReLU neural networks yield rate-distortion optimal approximation (Bölcskei et al., 2018) of a wide class of functions, including polynomials, windowed sinusoidal functions, one-dimensional oscillatory textures, and the Weierstrass function, a fractal function which is continuous but nowhere differentiable. Together with the recently established universal approximation result for affine function systems (Bölcskei et al., 2018), this demonstrates that deep neural networks approximate vastly different signal structures generated by the affine group, the Weyl-Heisenberg group, or through warping, and even certain fractals, all with approximation error decaying exponentially in the number of neurons. We also prove that in the approximation of sufficiently smooth functions finite-width deep networks require strictly fewer neurons than finite-depth wide networks.

## 1 INTRODUCTION

A theory establishing a link between the complexity of a neural network and the complexity of the function class to be approximated by the network was recently developed in Bölcskei et al. (2018). Based on this framework, it was shown (Bölcskei et al., 2018) that all affine function classes are optimally representable by neural networks in the sense of Kolmogorov rate-distortion theory (Donoho, 1993; Grohs, 2015). Equivalently, this means that the approximation error decays exponentially in the number of neurons employed in the approximation.

The present paper explores the question of whether the universality result established in Bölcskei et al. (2018) extends beyond affine function classes and answers it in the affirmative. Specifically, we consider the approximation of polynomials, windowed sinusoidal functions (Gröchenig & Samarah, 2000; Gröchenig, 2001), one-dimensional oscillatory textures according to Demanet & Ying (2007), and the Weierstrass function, a fractal function which is continuous everywhere and differentiable nowhere. The central conclusion of this paper is that finite-width ReLU networks of depth scaling poly-logarithmically in the inverse of the approximation error lead to exponentially decaying approximation error for all these different signal structures. This result is established by building on a recent breakthrough in Yarotsky (2016) and recognizing that the width of networks approximating polynomials need not scale linearly in the degree of the polynomial, but can actually be finite independently of the degree. This insight will also allow a sharp statement on the benefit of depth; specifically, we prove that in the approximation of sufficiently smooth functions finite-width deep networks require strictly fewer neurons than finite-depth wide networks.

**Notation.** For the function $f(x)\colon \mathbb{R}^d \to \mathbb{R}$, we define $\|f\|_{L^\infty(\Omega)} := \inf\{C \geq 0 : |f(x)| \leq C,$ for all $x \in \Omega\}$. For a vector $b \in \mathbb{R}^d$, we let $\|b\|_\infty := \max_{i=1,\dots,d} |b_i|$, similarly we write $\|A\|_\infty := \max_{i,j} |A_{i,j}|$ for the matrix $A \in \mathbb{R}^{m \times n}$. We denote the identity matrix of size $n \times n$ by $\mathbb{I}_n$. Throughout, $\log$ stands for the logarithm to base 2.

## 2 SETUP AND BASIC ReLU CALCULUS

We start by defining ReLU neural networks.

**Definition 2.1.** *Let $L, N_0, N_1, \ldots, N_L \in \mathbb{N}$. A map $\Phi : \mathbb{R}^{N_0} \to \mathbb{R}^{N_L}$ given by*

$$\Phi(x) = \begin{cases} W_2(\rho(W_1(x))), & L = 2 \\ W_L(\rho(W_{L-1}(\rho(\ldots \rho(W_1(x)))))), & L \geq 3 \end{cases}, \tag{1}$$

*with affine linear maps $W_\ell : \mathbb{R}^{N_{\ell-1}} \to \mathbb{R}^{N_\ell}$, $\ell \in \{1, 2, \ldots, L\}$, and the ReLU activation function $\rho(x) = \max(x, 0)$, $x \in \mathbb{R}$, acting component-wise (i.e., $\rho(x_1, \ldots, x_N) := (\rho(x_1), \ldots, \rho(x_N))$) is called a ReLU neural network. The map $W_\ell$ corresponding to layer $\ell$ is given by $W_\ell(x) = A_\ell x + b_\ell$, with $A_\ell \in \mathbb{R}^{N_\ell \times N_{\ell-1}}$ and $b_\ell \in \mathbb{R}^{N_\ell}$. We define the* network connectivity *as the total number of non-zero entries in the matrices $A_\ell$, $\ell \in \{1, 2, \ldots, L\}$, and the vectors $b_\ell$, $\ell \in \{1, 2, \ldots, L\}$. The depth of the network or, equivalently, the number of layers is $\mathcal{L}(\Phi) := L$ and its width $\mathcal{W}(\Phi) := \max_{\ell=0,\ldots,L} N_\ell$. We further denote by $\mathcal{B}(\Phi) := \max_{\ell=1,\ldots,L} \max\{\|A_\ell\|_\infty, \|b_\ell\|_\infty\}$ the maximum absolute value of the weights in the network.*

We designate the class of ReLU networks $\Phi : \mathbb{R}^d \to \mathbb{R}^{N_L}$ with no more than $L$ layers, width no more than $M$, input dimension $d$, and output dimension $N_L$ by $\mathcal{NN}_{L,M,d,N_L}$. Note that the connectivity of $\Phi$ is upper-bounded by $LM(M+1)$.

For later use we record three technical results. We first record a technical lemma on the composition of neural networks.[1]

**Lemma 2.2.** *Let $L_1, L_2, M_1, M_2, d_1, d_2, N_{L_1}, N_{L_2} \in \mathbb{N}$, $\Phi_1 \in \mathcal{NN}_{L_1,M_1,d_1,N_{L_1}}$, and $\Phi_2 \in \mathcal{NN}_{L_2,M_2,d_2,N_{L_2}}$ with $N_{L_1} = d_2$. Then, there exists a network $\Psi \in \mathcal{NN}_{L_1+L_2,\max\{2N_{L_1},M_1,M_2\},d_1,N_{L_2}}$ with $\mathcal{B}(\Psi) = \max\{\mathcal{B}(\Phi_1), \mathcal{B}(\Phi_2)\}$, satisfying $\Psi(x) = \Phi_2(\Phi_1(x))$, for all $x \in \mathbb{R}^{d_1}$.*

Before we can formalize the concept of a linear combination of neural networks, we need a result that shows how to augment network depth while retaining the networks input-output relation

**Lemma 2.3.** *Let $L, M, K, d \in \mathbb{N}$, $\Phi_1 \in \mathcal{NN}_{L,M,d,1}$, and $K > L$. Then, there exists a corresponding network $\Phi_2 \in \mathcal{NN}_{K,\max\{2,M\},d,1}$ such that $\Phi_2(x) = \Phi_1(x)$ for all $x \in \mathbb{R}^d$. Moreover, the weights of $\Phi_2$ consist of the weights of $\Phi_1$ and $\pm 1$'s.*

The next result formalizes the concept of a linear combination of neural networks.

**Lemma 2.4.** *Let $N, L_i, M_i, d_i \in \mathbb{N}$, $a_i \in \mathbb{R}$, $\Phi_i \in \mathcal{NN}_{L_i,M_i,d_i,1}$, $i = 1, 2, \ldots, N$, $d = \sum_{i=1}^N d_i$. Then, there exist networks $\Phi^1 \in \mathcal{NN}_{L,M,d,N}$ and $\Phi^2 \in \mathcal{NN}_{L,M,d,1}$ with $L = \max_i L_i$, and $M \leq \sum_{i=1}^N \max\{2, M_i\}$ satisfying*

$$\Phi^1(x) = (\Phi_1(x_1) \quad \Phi_2(x_2) \quad \ldots \quad \Phi_N(x_N))^T \quad \text{and}$$

$$\Phi^2(x) = \sum_{i=1}^N a_i \Phi_i(x_i),$$

*for all $x = (x_1^T, x_2^T, \ldots, x_N^T)^T \in \mathbb{R}^d$ with $x_i \in \mathbb{R}^{d_i}$, $i = 1, 2, \ldots, N$. Moreover, the weights of $\Phi^1$ consist of the weights of the networks $\Phi_i$, $i = 1, 2, \ldots, N$, and $\pm 1$'s. The weights of $\Phi^2$ consist of the weights of $\Phi^1$ and $\{a_1, a_2, \ldots, a_N\}$.*

**Remark 2.5.** *Note that if the networks $\Phi_i$ in Lemma 2.4 have shared inputs, the resulting networks $\Phi^1$ and $\Phi^2$ will have fewer than $d = \sum_{i=1}^N d_i$ inputs.*

## 3 APPROXIMATION OF POLYNOMIALS

This section shows how the multiplication operation and polynomials can be approximated to within error $\epsilon$ with ReLU networks of finite width and of depth poly-logarithmic in $1/\epsilon$. We also note that the approximation results throughout the paper guarantee that the magnitude of the weights in the network does not grow faster than polynomially in the size of the domain over which approximation takes place. Although not shown here for space constraints, the combination of finite width, depth

---

[1]The proofs of the following two lemmata, along with all other proofs not provided in the paper, can be found in the supplement.

scaling poly-logarithmically in $1/\epsilon$, and weights growing no faster than polynomially guarantees rate-distortion optimality in the sense of Bölcskei et al. (2018) and hence exponential error decay. Previous results on the approximation of the multiplication operation and of polynomials through finite-width ReLU networks reported in Hanin & Sellke (2017); Yarotsky (2018) do not come with bounds on the network weights, have depth scaling polynomially in $1/\epsilon$, and therefore do not allow to conclude rate-distortion optimality.

The proof ideas for the results in this section are inspired by Yarotsky (2016) and by the "sawtooth" construction of Telgarsky (2015). In contrast to Yarotsky (2016), we consider networks without "skip connections" and of finite and explicitly specified width. Before starting with the approximation of $x^2$, we note that all our results apply to the multivariate case as well, but we restrict ourselves to the univariate case for simplicity of exposition.

**Proposition 3.1.** *There exists a constant $C > 0$ such that for all $\epsilon \in (0, 1/2)$ there is a network $\Phi_\epsilon \in \mathcal{NN}_{\infty,4,1,1}$ satisfying $\mathcal{L}(\Phi_\epsilon) \leq C \log(\epsilon^{-1})$, $\mathcal{B}(\Phi_\epsilon) \leq 4$, $\Phi_\epsilon(0) = 0$, and*

$$\|\Phi_\epsilon(x) - x^2\|_{L^\infty([0,1])} \leq \epsilon. \tag{2}$$

With Proposition 3.1 we are now ready to show how ReLU networks can approximate the multiplication operation, which will then lead us to the approximation of arbitrary powers of $x$.

**Proposition 3.2.** *There exists a constant $C > 0$ such that for all $D \in \mathbb{R}_+$ and $\epsilon \in (0, 1/2)$ there is a network $\Phi_{D,\epsilon} \in \mathcal{NN}_{\infty,12,2,1}$ satisfying $\mathcal{L}(\Phi_{D,\epsilon}) \leq C \log(\lceil D \rceil^2 \epsilon^{-1})$, $\mathcal{B}(\Phi_{D,\epsilon}) \leq \max\{4, 2\lceil D \rceil^2\}$,*

$$\|\Phi_{D,\epsilon}(x, y) - xy\|_{L^\infty([-D,D]^2)} \leq \epsilon, \tag{3}$$

*and $\Phi_{D,\epsilon}(0, x) = \Phi_{D,\epsilon}(x, 0) = 0$, for all $x \in \mathbb{R}$.*

The next result establishes that arbitrary polynomials can be approximated by ReLU networks of finite and explicitly specified width and of depth growing logarithmically in the inverse of the approximation error. In particular, the width of the approximating network does not grow with the degree of the polynomial as is the case in Yarotsky (2016), Ding et al. (2018), Liang & Srikant (2017). This finite-width aspect is central to the approximation of sinusoidal functions by ReLU networks as described in the next section.

**Proposition 3.3.** *There exists a constant $C > 0$ such that for all $m \in \mathbb{N}$, $A \in \mathbb{R}_+$, $p_m(x) = \sum_{i=0}^m a_i x^i$ with $\max_{i=0,\dots,m} |a_i| = A$, $D \in \mathbb{R}_+$, and $\epsilon \in (0, 1/2)$, there is a network $\Phi_{p_m,D,\epsilon} \in \mathcal{NN}_{\infty,16,1,1}$ satisfying $\mathcal{L}(\Phi_{p_m,D,\epsilon}) \leq Cm(\log(\lceil A \rceil) + \log(\epsilon^{-1}) + m\log(\lceil D \rceil) + \log(m))$, $\mathcal{B}(\Phi_{p_m,D,\epsilon}) \leq \max\{A, 8\lceil D \rceil^{2m-2}\}$, and*

$$\|\Phi_{p_m,D,\epsilon} - p_m\|_{L^\infty([-D,D])} \leq \epsilon. \tag{4}$$

*Proof.* We start by noting that for $m = 1$ the resulting affine function $p_1(x) = a_0 + a_1 x$ can be realized exactly, i.e., with $\epsilon = 0$, by a network of depth $L = 2$ with

$$W_1(x) = \begin{pmatrix} a_1 \\ -a_1 \end{pmatrix} x + \begin{pmatrix} a_0 \\ -a_0 \end{pmatrix}$$

and $A_2 = (1 \ -1), b_2 = 0$. The proof for $m \geq 2$ will be effected by realizing the monomials $x^k, k \geq 2$, through iterative composition of multiplication networks and combining this with a construction which uses the network realizing $x^k$ not only as a building block in the network realizing $x^{k+1}$ but also to construct the network approximating the partial sum $\sum_{i=0}^k a_i x^i$ in parallel.

We start by setting $H_{D,\eta}^k := \lceil D \rceil^k + \eta \sum_{s=0}^{k-2} \lceil D \rceil^s$, $k \in \mathbb{N}$, and let $\Phi_{H_{D,\eta}^k, \eta}$, $D \in \mathbb{R}_+$, $k \in \mathbb{N}$, $\eta \in (0, 1/2)$, be multiplication networks according to Proposition 3.2. For $D \in \mathbb{R}_+$, $k \in \mathbb{N}$, $\eta \in (0, 1/2)$, we then recursively define $\Psi_{D,\eta}^k$ according to $\Psi_{D,\eta}^0(x) = 1$, $\Psi_{D,\eta}^1(x) = x$, and $\Psi_{D,\eta}^k(x) = \Phi_{H_{D,\eta}^{k-1}, \eta}(x, \Psi_{D,\eta}^{k-1}(x)), k \geq 2$. Note that $\Psi_{D,\eta}^k(x)$ can be realized through a neural network for all $k \in \mathbb{N}$ thanks to Lemma 2.2 and the fact that, as already noted above for the case $m = 1$, any affine function can be realized through a neural network.

We first show by induction that

$$\|\Psi_{D,\eta}^k(x) - x^k\|_{L^\infty([-D,D])} \leq \eta \sum_{s=0}^{k-2} \lceil D \rceil^s, \tag{5}$$

for all $\eta \in (0, 1/2)$, $k \geq 2$. The base case $k = 2$ follows from

$$\|\Psi_{D,\eta}^2(x) - x^2\|_{L^\infty([-D,D])} = \|\Phi_{H_{D,\eta}^1,\eta}(x,x) - x^2\|_{L^\infty([-D,D])} \leq \eta.$$

We proceed to establishing the induction step $(k-1) \to k$. The induction assumption is

$$\|\Psi_{D,\eta}^{k-1}(x) - x^{k-1}\|_{L^\infty([-D,D])} \leq \eta \sum_{s=0}^{k-3} \lceil D \rceil^s. \tag{6}$$

Since $\|\Psi_{D,\eta}^{k-1}\|_{L^\infty([-D,D])} \leq \|x^{k-1}\|_{L^\infty([-D,D])} + \|\Psi_{D,\eta}^{k-1}(x) - x^{k-1}\|_{L^\infty([-D,D])} \leq H_{D,\eta}^{k-1}$, Proposition 3.2 implies that

$$\|\Psi_{D,\eta}^k(x) - x^k\|_{L^\infty([-D,D])} \leq \|\Phi_{H_{D,\eta}^{k-1},\eta}(x, \Psi_{D,\eta}^{k-1}(x)) - x\Psi_{D,\eta}^{k-1}(x)\|_{L^\infty([-D,D])}$$
$$+ \max_{[-D,D]} |x| \|\Psi_{D,\eta}^{k-1}(x) - x^{k-1}\|_{L^\infty([-D,D])}$$
$$\leq \eta + \lceil D \rceil \eta \sum_{s=0}^{k-3} \lceil D \rceil^s = \eta \sum_{s=0}^{k-2} \lceil D \rceil^s,$$

which completes the proof of the induction step.

We are now ready to proceed to the construction of the network $\Phi_{p_m,D,\epsilon}$ approximating the polynomial $p_m(x) = \sum_{i=0}^m a_i x^i$. To this end, we first note that the identity mapping $x \mapsto x$ and the linear combination $x, y \mapsto x + a_{i-1}y$ are affine transformations and can thus be realized by a network of depth $L = 2$. By Lemma 2.4 there hence exists a constant $C_2$ such that for every $m \geq 2$, $p_m(x) = \sum_{\ell=0}^m a_\ell x^\ell$, $i \in \{2, 3, \ldots, m\}$, $\eta \in (0, 1/2)$ there is a network $\varphi_{p_m,D,\eta}^i \in \mathcal{NN}_{\infty,16,3,3}$ with $\mathcal{L}(\varphi_{p_m,D,\eta}^i) \leq C_2 \log(\lceil H_{D,\eta}^{i-1} \rceil^2 \eta^{-1})$, and $\mathcal{B}(\varphi_{p_m,D,\eta}^i) \leq \max\{4, 2\lceil H_{D,\eta}^{i-1} \rceil^2, \max_{i \in \{0,\ldots,m\}} |a_i|\}$ realizing the map

$$(x \quad s \quad y)^\top \to \left(x \quad s + a_{i-1}y \quad \Phi_{H_{D,\eta}^{i-1},\eta}(x,y)\right)^\top.$$

The statements in the following apply for all $m \in \mathbb{N}$, $A \in \mathbb{R}_+$, $p_m(x) = \sum_{i=0}^m a_i x^i$ with $\max_{i=0,\ldots,m} |a_i| \leq A$, $D \in \mathbb{R}_+$, and $\epsilon \in (0, 1/2)$. The network $\Phi_{p_m,D,\epsilon}$ approximating the polynomial $p_m(x) = \sum_{i=0}^m a_i x^i$ is now constructed according to

$$\Phi_{p_m,D,\epsilon}(x) := (0 \quad 1 \quad a_m) \varphi_{p_m,D,\eta_\epsilon}^m \left(\varphi_{p_m,D,\eta_\epsilon}^{m-1}\left(\cdots \varphi_{p_m,D,\eta_\epsilon}^2\left(\begin{pmatrix}1\\0\\1\end{pmatrix}x + \begin{pmatrix}0\\a_0\\0\end{pmatrix}\right)\right)\right),$$

with $\eta_\epsilon := (\lceil A \rceil m^2 \lceil D \rceil^m)^{-1}\epsilon$. This yields

$$\Phi_{p_m,D,\epsilon}(x) = \sum_{i=0}^m a_i \Psi_{D,\eta_\epsilon}^i(x), \quad \text{for all } x \in \mathbb{R}.$$

Hence equation 5 implies

$$\left\|\Phi_{p_m,D,\epsilon}(x) - p_m\right\|_{L^\infty([-D,D])} \leq \sum_{i=0}^m |a_i| \|\Psi_{D,\eta_\epsilon}^i(x) - x^i\|_{L^\infty([-D,D])} \leq \sum_{i=2}^m |a_i|\left(\eta_\epsilon \sum_{s=0}^{i-2} \lceil D \rceil^s\right)$$
$$\leq \eta_\epsilon \max_{i \in \{2,\ldots,m\}} |a_i| \sum_{i=2}^m (i-1)\lceil D \rceil^{i-2} \leq Am^2 \lceil D \rceil^{m-2}\eta_\epsilon \leq \epsilon.$$

Thanks to its compositional structure, the width of $\Phi_{p_m,D,\epsilon}$ equals the maximum width of the individual networks in the composition, i.e., $\mathcal{W}(\Phi_{p_m,D,\epsilon}) \leq 16$. Since $H_{D,\eta_\epsilon}^{i-1} \leq 2\lceil D \rceil^{m-1}$, for $i \leq m$, we further have

$$\mathcal{L}(\Phi_{p_m,D,\epsilon}) \leq \sum_{i=2}^m \mathcal{L}(\varphi_{p_m,D,\eta_\epsilon}^i) \leq \sum_{i=2}^m C_2 \log(\lceil H_{D,\eta_\epsilon}^{i-1} \rceil^2 \eta_\epsilon^{-1})$$
$$\leq C_2 m \left(\log(\lceil A \rceil) + \log(\epsilon^{-1}) + (3m-2)\log(\lceil D \rceil) + 2\log(m) + 2\right)$$
$$\leq 4 C_2 m \left(\log(\lceil A \rceil) + \log(\epsilon^{-1}) + m\log(\lceil D \rceil) + \log(m)\right).$$

Finally, we note that

$$\mathcal{B}(\Phi_{p_m,D,\epsilon}) = \max\{1, |a_0|, |a_m|, \max_{i \in \{2,3,\dots,m\}} \mathcal{B}(\varphi^i_{p_m,D,\eta_\epsilon})\} \leq \max\{A, 8\lceil D \rceil^{2m-2}\}.$$

This finalizes the proof. $\qquad \square$

We conclude this section with a result on ReLU networks approximating smooth functions with exponential accuracy. The proof of this statement, provided in the supplement, is based on the theory developed above.

**Definition 3.4.** *For $D \in \mathbb{R}_+$, let the set $\mathcal{S}_D \subseteq C^\infty([-D, D], \mathbb{R})$ be given by*

$$\mathcal{S}_D = \left\{ f \in C^\infty([-D, D], \mathbb{R}) \colon \|f^{(n)}(x)\|_{L^\infty([-D,D])} \leq n!, \text{ for all } n \in \mathbb{N}_0 \right\}. \tag{7}$$

**Lemma 3.5.** *There exist constants $C > 0$ and a polynomial $\pi$ such that for all $D \in \mathbb{R}_+$, $f \in \mathcal{S}_D$, and $\epsilon \in (0, 1/2)$, there is a network $\Psi_{f,\epsilon} \in \mathcal{NN}_{\infty,23,1,1}$ satisfying $\mathcal{L}(\Psi_{f,\epsilon}) \leq C\lceil D \rceil (\log(\epsilon^{-1}))^2$, $\mathcal{B}(\Psi_{f,\epsilon}) \leq \max\{1/D, \lceil D \rceil\}\pi(\epsilon^{-1})$, and*

$$\|\Psi_{f,\epsilon} - f\|_{L^\infty([-D,D])} \leq \epsilon. \tag{8}$$

Note that for expositional simplicity Lemma 3.5 covers functions defined on symmetric intervals $[-D, D]$. Close inspection of the proof reveals, however, that for arbitrary intervals $[a, b]$ and functions $f \in C^\infty([a, b], \mathbb{R})$ with $\|f^{(n)}\|_{L^\infty([a,b])} \leq n!$, for all $n \in \mathbb{N}$, the same statement (with $D$ replaced by $b - a$ in the bounds on $\mathcal{L}(\Psi_{f,\epsilon}), \mathcal{B}(\Psi_{f,\epsilon})$) holds.

## 4 APPROXIMATION OF SINUSOIDAL FUNCTIONS

We are now ready to proceed to the approximation of sinusoidal functions.

**Theorem 4.1.** *There exists a constant $C$ such that for every $a, D \in \mathbb{R}_+$, $\epsilon \in (0, 1/2)$, there is a network $\Psi_{a,D,\epsilon} \in \mathcal{NN}_{\infty,16,1,1}$ satisfying $\mathcal{L}(\Psi_{a,D,\epsilon}) \leq C((\log(1/\epsilon))^2 + \log(\lceil aD \rceil))$, $\mathcal{B}(\Psi_{a,D,\epsilon}) \leq C$, and*

$$\|\Psi_{a,D,\epsilon} - \cos(a \cdot)\|_{L^\infty([-D,D])} \leq \epsilon.$$

*Proof.* We start by approximating $x \mapsto \cos(2\pi x)$ on $[0, 1]$. To this end note the MacLaurin series representation

$$\cos(x) = \sum_{n=0}^\infty \frac{(-1)^n}{(2n)!} x^{2n}, \quad \forall x \in \mathbb{R}.$$

Thanks to the Taylor theorem with remainder in Lagrange form, we have, for all $x \in [0, 1]$,

$$\left| \cos(2\pi x) - \sum_{n=0}^N \frac{(-1)^n}{(2n)!} (2\pi x)^{2n} \right| \leq \left| \frac{(2\pi x)^{2N+1}}{(2N+1)!} \right| \sup_{t \in [0,1]} |\cos^{(2N+1)}(2\pi t)| \leq \frac{(2\pi)^{4N+2}}{(2N+1)!}. \tag{9}$$

Next observe that $n! \geq (\frac{n}{e})^n e$, for all $n \in \mathbb{N}$, which implies,

$$\frac{(2\pi)^{4N+2}}{(2N+1)!} \leq \frac{(4\pi^2)^{2N+1}}{(\frac{2N+1}{e})^{2N+1} e} \leq \left( \frac{4\pi^2 e}{2N+1} \right)^{2N+1}, \quad \text{for all } N \in \mathbb{N}. \tag{10}$$

With $N_\epsilon := \lceil 2\pi^2 e \log(2/\epsilon) \rceil$, we get, for all $\epsilon \in (0, 1/2)$,

$$\left( \frac{4\pi^2 e}{2N_\epsilon + 1} \right)^{2N_\epsilon + 1} = \left( \frac{4\pi^2 e}{2\lceil 2\pi^2 e \log(2/\epsilon) \rceil + 1} \right)^{2\lceil 2\pi^2 e \log(2/\epsilon) \rceil + 1} \leq 2^{-\lceil 2\pi^2 e \log(2/\epsilon) \rceil}$$

$$\leq 2^{-\log(2/\epsilon)} = \frac{\epsilon}{2}. \tag{11}$$

Noting that $C_1 := \left\lceil \max_{n \in \mathbb{N}_0} \left( \frac{(2\pi)^{2n}}{(2n)!} \right) \right\rceil < \infty$ and $N_\epsilon \leq C_2 \log(\epsilon^{-1})$, for all $\epsilon \in (0, 1/2)$, with $C_2 := 4\pi^2 e + 1$, application of Proposition 3.3 to

$$p_m(x) = p_{N_\epsilon}(x) := \sum_{n=0}^{N_\epsilon} \frac{(-1)^n}{(2n)!} (2\pi x)^{2n},$$

with $D = 1$, establishes the following: There is a constant $C_3$ such that, for all $\epsilon \in (0, 1/2)$, there is a network $\Phi_{\epsilon/2}$ satisfying

$$\left\| \Phi_{\epsilon/2} - p_{N_\epsilon} \right\|_{L^\infty([-1,1])} \leq \frac{\epsilon}{2}, \tag{12}$$

with $\mathcal{W}(\Phi_{\epsilon/2}) \leq 16$, $\mathcal{B}(\Phi_{\epsilon/2}) \leq C_3$, and

$$\mathcal{L}(\Phi_{\epsilon/2}) \leq C_3 N_\epsilon (\log(C_1) + \log(2/\epsilon) + N_\epsilon \log(1) + \log(N_\epsilon)) \leq C_4 (\log(\epsilon^{-1}))^2,$$

where $C_4 := C_2 C_3 (1 + C_1 + C_2)$. Combining equation 9, equation 10, equation 11, and equation 12, it follows that the network $\Phi_{\epsilon/2}$ approximates the function $x \mapsto \cos(2\pi x)$ on $[0, 1]$ to within accuracy $\epsilon$, i.e., for all $\epsilon \in (0, 1/2)$, we have

$$\|\Phi_{\epsilon/2} - \cos(2\pi \cdot)\|_{L^\infty([0,1])} \leq \epsilon. \tag{13}$$

We next extend this result to the approximation of $x \mapsto \cos(ax)$ on the interval $[-1, 1]$ for arbitrary $a \in \mathbb{R}_+$. This will be accomplished by exploiting that $x \mapsto \cos(2\pi x)$ is 1-periodic and even. First recall the "sawtooth" functions $g_s \colon [0, 1] \to [0, 1]$, $s \in \mathbb{N}$, as defined in equation 24. It is straightforward, albeit somewhat tedious, to see that, for all $s \in \mathbb{N}_0$, $x \in [0, 1]$,

$$\cos(2\pi 2^s x) = \cos(2\pi g_s(x)).$$

Similarly, it follows that $\cos(2\pi 2^s x) = \cos(2\pi g_s(|x|))$, for all $s \in \mathbb{N}_0$, $x \in [-1, 1]$. Next, note that for every $a \in \mathbb{R}_+$, there exists a $C_a \in (1/2, 1]$ such that $a/(2\pi) = C_a 2^{\lceil \log(a) - \log(2\pi) \rceil}$; we thus have, for all $a \in \mathbb{R}_+$, $x \in [-1, 1]$,

$$\cos(ax) = \cos(2\pi 2^{\lceil \log(a) - \log(2\pi) \rceil} C_a x) = \cos\left(2\pi g_{\lceil \log(a) - \log(2\pi) \rceil}(C_a |x|)\right).$$

Since $g_{\lceil \log(a) - \log(2\pi) \rceil}(C_a |x|) \in [0, 1]$, for all $a \in \mathbb{R}_+$, $x \in [-1, 1]$, it follows from equation 13 that

$$\begin{aligned}
&\left\| \Phi_{\epsilon/2}\left(g_{\lceil \log(a) - \log(2\pi) \rceil}(C_a |x|)\right) - \cos\left(2\pi g_{\lceil \log(a) - \log(2\pi) \rceil}(C_a |x|)\right) \right\|_{L^\infty([-1,1])} \\
&= \left\| \Phi_{\epsilon/2}\left(g_{\lceil \log(a) - \log(2\pi) \rceil}(C_a |x|)\right) - \cos(ax) \right\|_{L^\infty([-1,1])} \leq \epsilon.
\end{aligned} \tag{14}$$

Now recall that $x \mapsto |x| = \rho(x) + \rho(-x)$ can be implemented by a 2-layer network and consider the realization of $x \mapsto g_{\lceil \log(a) - \log(2\pi) \rceil}(C_a x)$, $a \in \mathbb{R}_+$, as developed in the proof of Proposition 3.1. Applying Lemma 2.2 twice, then establishes, thanks to (14), the existence of a constant $C_5$ such that the network

$$\Psi_{a,\epsilon} := \Phi_{\epsilon/2}(g_{\lceil \log(a) - \log(2\pi) \rceil}(C_a |x|))$$

approximates $x \mapsto \cos(ax)$ on $[-1, 1]$ with accuracy $\epsilon$, while satisfying $\mathcal{L}(\Psi_{a,\epsilon}) \leq C_5((\log(1/\epsilon))^2 + \log(\lceil a \rceil))$, $\mathcal{W}(\Psi_{a,\epsilon}) \leq 16$, and $\mathcal{B}(\Psi_{a,\epsilon}) \leq C_5$.

Finally, we consider the approximation of $x \mapsto \cos(ax)$ on intervals $[-D, D]$, for arbitrary $D \geq 1$. To this end, we define, for all $a \in \mathbb{R}_+$, $D \in [1, \infty)$, $\epsilon \in (0, 1/2)$, the network $\Psi_{a,D,\epsilon}(x) := \Psi_{aD,\epsilon}(\frac{x}{D})$ and observe that

$$\begin{aligned}
\sup_{x \in [-D,D]} |\Psi_{a,D,\epsilon}(x) - \cos(ax)| &= \sup_{y \in [-1,1]} |\Psi_{a,D,\epsilon}(Dy) - \cos(aDy)| \\
&= \sup_{y \in [-1,1]} |\Psi_{aD,\epsilon}(y) - \cos(aDy)| \leq \epsilon.
\end{aligned}$$

This concludes the proof. $\qquad\square$

An approximation result for $\sin(ax)$ can be obtained directly from Theorem 4.1 simply by noting that $\sin(x) = \cos(x - \pi/2)$, which can be realized by the concatenation of a neural network that performs an affine transformation and a network that approximates $\cos(x)$. The formal statement is as follows.

**Corollary 4.2.** *There exists a constant $C > 0$ such that for every $a, D \in \mathbb{R}_+$, $b \in \mathbb{R}$, $\epsilon \in (0, 1/2)$ there is a network $\Psi_{a,b,D,\epsilon} \in \mathcal{NN}_{\infty,16,1,1}$ satisfying*

$$\|\Psi_{a,b,D,\epsilon} - \cos(a \cdot - b)\|_{L^\infty([-D,D])} \leq \epsilon, \tag{15}$$

*with $\mathcal{L}(\Psi_{a,b,D,\epsilon}) \leq C((\log(\epsilon^{-1}))^2 + \log(\lceil aD + |b| \rceil))$ and $\mathcal{B}(\Psi_{a,b,D,\epsilon}) \leq C$.*

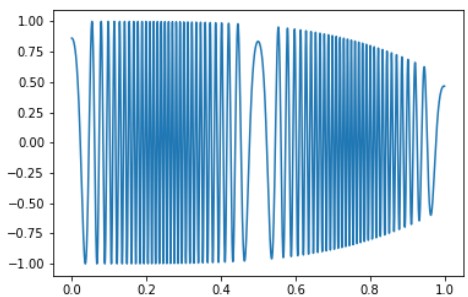 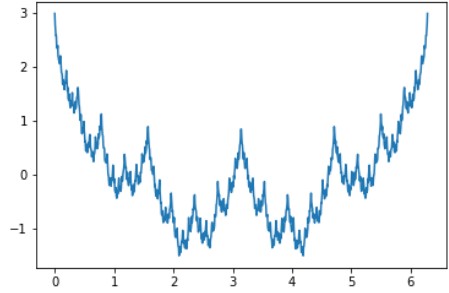

Figure 1: Left: A function in $\mathcal{F}_{1,100}$. Right: The function $W_{\frac{1}{\sqrt{2}},2}$.

## 5 OSCILLATORY TEXTURES AND THE WEIERSTRASS FUNCTION

Consider the following function class consisting of one-dimensional "oscillatory textures" according to Demanet & Ying (2007).

**Definition 5.1.** *Let the sets $\mathcal{F}_{D,a}$, $D, a \in \mathbb{R}_+$, be given by*

$$\mathcal{F}_{D,a} = \{\cos(ag)h \colon g, h \in \mathcal{S}_D\}. \tag{16}$$

The efficient approximation of functions in $\mathcal{F}_{D,a}$ with $a$ large is a notoriously difficult problem due to the combination of the rapidly oscillating cosine term and the warping $g$. The best available approximation results in the literature (Demanet & Ying, 2007) are based on wave-atom dictionaries[2] and yield low-order polynomial approximation rates. In what follows we show that finite-width deep networks drastically improve these results to exponential approximation rates.

**Proposition 5.2.** *There exist a constant $C > 0$ and a polynomial $\pi$ such that for all $D, a \in \mathbb{R}_+$, $f \in \mathcal{F}_{D,a}$, and $\epsilon \in (0, 1/2)$ there is a network $\Gamma_{f,\epsilon} \in \mathcal{NN}_{\infty,46,1,1}$ which satisfies*

$$\|\Gamma_{f,\epsilon} - f\|_{L^\infty([-D,D])} \le \epsilon, \tag{17}$$

*with $\mathcal{L}(\Gamma_{f,\epsilon}) \le C(\lceil D \rceil (\log(\epsilon^{-1}))^2 + \log(\lceil aD \rceil))$ and $\mathcal{B}(\Gamma_{f,\epsilon}) \le \max\{1/D, \pi(\epsilon^{-1}, \lceil D \rceil, \lceil a \rceil)\}$.*

Finally, we show how the Weierstrass function—a fractal function, which is continuous everywhere but differentiable nowhere—can be approximated with exponential accuracy by deep ReLU networks. Specifically, we consider

$$W_{p,a}(x) = \sum_{k=0}^{\infty} p^k \cos(a^k \pi x), \quad \text{for } p \in (0, 1/2], \, a \in \mathbb{R}_+. \tag{18}$$

Let $\alpha = -\frac{\log(p)}{\log(a)}$. It is well known (Zygmund, 2002) that $W_{p,a}$ possesses Hölder smoothness $\alpha$ which may be arbitrarily small, depending on $p$ and $a$, see Figure 1 right. While classical approximation methods, for instance sparse approximation in frames, are not suitable owing to the warping operation, it turns out that deep finite-width networks achieve exponential approximation rate. The corresponding formal statement is as follows.

**Proposition 5.3.** *There exists a constant $C > 0$ such that, for all $\epsilon \in (0, 1/2)$, $p \in (0, 1/2]$, $a \in \mathbb{R}_+$, $D \ge 1$, there is a network $\Psi_{p,a,D,\epsilon} \in \mathcal{NN}_{\infty,20,1,1}$ satisfying*

$$\|\Psi_{p,a,D,\epsilon} - W_{p,a}\|_{L^\infty([-D,D])} \le \epsilon, \tag{19}$$

*with $\mathcal{L}(\Psi_{p,a,D,\epsilon}) \le C((\log(1/\epsilon))^3 + 2(\log(1/\epsilon))^2 \log(\lceil a \rceil) + \log(1/\epsilon) \log(D))$ and $\mathcal{B}(\Psi_{p,a,D,\epsilon}) \le C$.*

---

[2]To be precise, the results in Demanet & Ying (2007) are concerned with the two-dimensional case, whereas we focus on the one-dimensional case. Note, however, that all our results can be readily extended to the multivariate case.

## 6 FINITE DEPTH IS NOT ENOUGH

We next show that, in the approximation of periodic functions, finite-width deep networks require asymptotically smaller connectivity than finite-depth wide networks. This statement is then extended to sufficiently smooth non-periodic functions, thereby formalizing the benefit of deep networks over shallow networks in the approximation of a broad class of functions.

We start with preparatory material taken from Telgarsky (2015).

**Definition 6.1** (Telgarsky (2015))**.** *Let $k \in \mathbb{N}$. A function $f : \mathbb{R} \to \mathbb{R}$ is called $k$-sawtooth if it is piecewise linear with no more than $k$ pieces, i.e., its domain $\mathbb{R}$ can be partitioned into $k$ intervals such that $f$ is linear on each of these intervals.*

**Lemma 6.2** (Telgarsky (2015))**.** *Every $\Phi \in \mathcal{NN}_{L,M,1,1}$ is $(2M)^L$-sawtooth.*

**Definition 6.3.** *For a non-constant $u$-periodic function $f \in C(\mathbb{R})$, we define*

$$\xi(f) := \inf_{\substack{\delta \in [0,u), \\ c,d \in \mathbb{R}}} \|f(x) - (cx + d)\|_{L^\infty([\delta, u+\delta])}.$$

The quantity $\xi(f)$ measures the error incurred by the best linear approximation of $f$ on any segment of length equal to the period of $f$; It can hence be interpreted as quantifying the non-linearity of $f$. The next result states that finite-depth networks with width scaling poly-logarithmically in the highest frequency of the periodic function to be approximated can not achieve arbitrarily small approximation error.

**Proposition 6.4.** *Let $f \in C(\mathbb{R})$ be a non-constant $u$-periodic function, let $L \in \mathbb{N}$ and $\pi$ a polynomial. Then there exists $a \in \mathbb{R}_+$ such that for every network $\Phi \in \mathcal{NN}_{L,M,1,1}$ with $M \leq \pi(\log(a))$ it holds that*

$$\|f(ax) - \Phi(x)\|_{L^\infty([0,u])} \geq \xi(f) > 0.$$

Application of Proposition 6.4 shows that finite-depth networks, owing to $\xi(\cos) > 0$, require faster than poly-logarithmic growth of connectivity in $a$ to approximate $x \mapsto \cos(ax)$ with arbitrarily small error, whereas finite-width networks, thanks to Theorem 4.1, can accomplish this with poly-logarithmic growth in connectivity. The next result, taken from (Frenzen et al., 2010), allows us to extend this conclusion to non-periodic sufficiently smooth functions.

**Theorem 6.5** (Frenzen et al. (2010))**.** *Let $f \in C^3([a,b])$ and consider a piecewise linear approximation of $f$ on $[a,b]$ that is accurate to within $\epsilon$ in the $L^\infty([a,b])$-norm. The minimal number of linear pieces required to accomplish this scales according to*

$$s(\epsilon) \sim \frac{c}{\sqrt{\epsilon}}, \epsilon \to 0, \text{ where } c = \frac{1}{4} \int_a^b \sqrt{|f''(x)|} dx.$$

Combining this with Lemma 6.2 yields the following statement on the depth-width tradeoff of networks approximating three-times continuously differentiable functions.

**Theorem 6.6.** *Let $f \in C^3([a,b])$ with $\int_a^b \sqrt{|f''(x)|} dx > 0$, $L \in \mathbb{N}$, and let $\pi$ be a polynomial. Then there exists $\epsilon > 0$ such that for every network $\Phi \in \mathcal{NN}_{L,M,1,1}$ with $M \leq \pi(\log(\epsilon^{-1}))$ it holds that*

$$\|f - \Phi\|_{L^\infty([a,b])} \geq \epsilon.$$

This shows that any function which is at least three times continuously differentiable cannot be approximated by finite-depth networks of connectivity scaling poly-logarithmically. In contrast, as Proposition 3.3 and Theorem 4.1 show, finite-width networks can approximate various interesting types of smooth functions such as polynomials and sinusoidal functions at poly-logarithmic connectivity growth rates. Further results on the limitations of finite-depth networks akin to Theorem 6.6 were reported recently in Petersen & Voigtlaender (2017).

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

# A  PROOFS

## A.1  PROOF OF LEMMA 2.2

*Proof.* The proof is based on the identity $x = \rho(x) - \rho(-x)$. First, note that by Definition 2.1, we can write

$$\Phi_1(x) = W_{L_1}^1(\rho(\ldots W_1^1(x))) \text{ and } \Phi_2(x) = W_{L_2}^2(\rho(\ldots W_1^2(x))).$$

Next, define the affine map given by $\widetilde{W}(x) = W_1^2\left(\begin{pmatrix}\mathbb{I}_{N_{L_1}} & -\mathbb{I}_{N_{L_1}}\end{pmatrix} x\right)$, for $x \in \mathbb{R}^{2N_{L_1}}$, and note that thanks to

$$W_1^2(\Phi_1(x)) = \widetilde{W}\left(\rho\left(\begin{pmatrix}W_{L_1}^1 \\ -W_{L_1}^1\end{pmatrix}(\rho(\ldots W_1^1(x)))\right)\right),$$

the map

$$\Psi(x) = W_{L_2}^2\left(\rho\left(\ldots W_2^2\left(\rho\left(\widetilde{W}\left(\rho\left(\begin{pmatrix}W_{L_1}^1 \\ -W_{L_1}^1\end{pmatrix}(\rho(\ldots W_1^1(x)))\right)\right)\right)\right)\right)\right)$$

satisfies $\Psi(x) = \Phi_2(\Phi_1(x))$, for all $x \in \mathbb{R}^{d_1}$, with $\mathcal{L}(\Psi) = L_1 + L_2$, $\mathcal{M}(\Psi) \le 2M_1 + 2M_2$, $\mathcal{W}(\Psi_1) \le \max\{2N_{L_1}, \mathcal{W}(\Phi_1), \mathcal{W}(\Phi_2)\}$, and $\mathcal{B}(\Psi) \le \max\{\mathcal{B}(\Phi_1), \mathcal{B}(\Phi_2)\}$. $\square$

## A.2  PROOF OF LEMMA 2.3

*Proof.* The proof is based on the identity $x = \rho(x) - \rho(-x)$. First, note that by equation 1 we can write $\Phi_1(x) = W_L(\rho(\ldots W_1(x)))$. For $K = L + 1$, $\Phi_2$ is given by

$$\Phi_2(x) = \begin{pmatrix}1 & -1\end{pmatrix}\rho\left(\begin{pmatrix}W_L \\ -W_L\end{pmatrix}(\rho(\ldots W_1(x)))\right) \in \mathcal{NN}_{L+1,\max\{2,M\},d,1}. \tag{20}$$

For $K > L + 1$, consider the network

$$\Phi_1'(x) = \begin{pmatrix}\rho(\Phi_1(x)) \\ \rho(-\Phi_1(x))\end{pmatrix} = \begin{pmatrix}1 & 0 \\ 0 & 1\end{pmatrix}\rho\left(\begin{pmatrix}W_L \\ -W_L\end{pmatrix}(\rho(\ldots W_1(x)))\right) \in \mathcal{NN}_{L+1,\max\{2,M\},d,2}, \tag{21}$$

which satisfies $\mathcal{W}(\Phi_1') = \max\{2, \mathcal{W}(\Phi_1)\}$. Next, we note that for every network of the form $\Psi(x) = \mathbb{I}_2\,\rho(\ldots)$, the network

$$\Psi'(x) := \mathbb{I}_2\,\rho(\Psi(x)), \tag{22}$$

satisfies $\Psi'(x) = \Psi(x)$, for all $x \in \mathbb{R}^d$, $\mathcal{L}(\Psi') = \mathcal{L}(\Psi) + 1$, and $\mathcal{W}(\Psi') = \max\{\mathcal{W}(\Psi), 2\}$. Moreover, the weights of $\Psi'$ consist of the weights of $\Psi$ and $\{1\}$. Noting that $\Phi_1'$ in (21) is of the form $\mathbb{I}_2\,\rho(\ldots)$ and iteratively applying the operation in equation 22 $K - L - 2$ times to $\Phi_1'$, we obtain a network $\Phi_1'' \in \mathcal{NN}_{K-1,\max\{2,M\},d,2}$. The proof is concluded by noting that $\Phi_2 = \begin{pmatrix}1 & -1\end{pmatrix}\rho(\Phi_1'') \in \mathcal{NN}_{K,\max\{2,M\},d,1}$ satisfies $\Phi_2(x) = \Phi_1(x)$, for all $x \in \mathbb{R}^d$. $\square$

## A.3  PROOF OF LEMMA 2.4

*Proof.* Apply Lemma 2.3 to the networks $\Phi_i$ to get corresponding networks $\tilde{\Phi}_i$ of depth $L$ and set $\Phi^1(x) := \left(\tilde{\Phi}_1(x_1), \tilde{\Phi}_2(x_2), \ldots, \tilde{\Phi}_N(x_N)\right)^\top$, $\Phi^2(x) := (a_1, a_2, \ldots, a_N)\Phi^1(x)$. $\square$

## A.4  PROOF OF PROPOSITION 3.1

*Proof.* Consider the function $g : [0, 1] \to [0, 1]$,

$$g(x) = \begin{cases}2x, & \text{if } x < \frac{1}{2}, \\ 2(1 - x), & \text{if } x \ge \frac{1}{2},\end{cases} \tag{23}$$

along with the "sawtooth" functions given by its $s$-fold composition

$$g_s := \underbrace{g \circ g \circ \cdots \circ g}_{s}, \quad s \ge 2, \tag{24}$$

and set $g_0(x) := x, g_1(x) := g(x)$. We next briefly review a fundamental result from Yarotsky (2016) showing how the function $f(x) := x^2, x \in [0, 1]$, can be approximated by linear combinations of "sawtooth" functions $g_s$. Specifically, let $f_m$ be the piecewise linear interpolation of $f$ with $2^m + 1$ uniformly spaced "knots" according to

$$f_m\left(\frac{k}{2^m}\right) = \left(\frac{k}{2^m}\right)^2, \quad k = 0, \ldots, 2^m, \quad m \in \mathbb{N}_0.$$

The function $f_m$ approximates $f$ with error $\epsilon_m = 2^{-2m-2}$ in the sense of

$$\|f_m(x) - x^2\|_{L^\infty[0,1]} \leq 2^{-2m-2}.$$

Next, note that we can refine interpolation in the sense of going from $f_{m-1}$ to $f_m$ by adjustment with a sawtooth function according to

$$f_m(x) = f_{m-1}(x) - \frac{g_m(x)}{2^{2m}}. \tag{25}$$

This leads to the representation

$$f_m(x) = x - \sum_{s=1}^{m} \frac{g_s(x)}{2^{2s}}. \tag{26}$$

While Yarotsky's construction Yarotsky (2016) is finalized by realizing equation 26 through a deep ReLU network of width 3 with the help of skip connections He et al. (2016), i.e., connections between nodes in non-consecutive layers, we proceed by constructing an equivalent (in terms of input-output relation) network without skip connections and of width 4. As $g(x) = 2\rho(x) - 4\rho(x - 1/2) + 2\rho(x - 1)$, it follows that

$$g_m = 2\rho(g_{m-1}) - 4\rho(g_{m-1} - 1/2) + 2\rho(g_{m-1} - 1), \tag{27}$$

and since $f_m = \rho(f_m), \forall m \in \mathbb{N}_0$, equation 25 can be rewritten as

$$f_m = \rho(f_{m-1}) - 2^{-2m}\Big(2\rho(g_{m-1}) - 4\rho(g_{m-1} - 1/2) + 2\rho(g_{m-1} - 1)\Big). \tag{28}$$

Equivalently, equation 27 and equation 28 can be cast as a composition of affine linear maps and a ReLU nonlinearity according to

$$\begin{pmatrix} g_m \\ f_m \end{pmatrix} = W_1\left(\rho\left(W_2\begin{pmatrix} g_{m-1} \\ f_{m-1} \end{pmatrix}\right)\right), \tag{29}$$

with

$$W_1(x) = \begin{pmatrix} 2 & -4 & 2 & 0 \\ -2^{-2m+1} & 2^{-2m+2} & -2^{-2m+1} & 1 \end{pmatrix} \begin{pmatrix} x_1 \\ x_2 \\ x_3 \\ x_4 \end{pmatrix}, \quad W_2(x) = \begin{pmatrix} 1 & 0 \\ 1 & 0 \\ 1 & 0 \\ 0 & 1 \end{pmatrix} \begin{pmatrix} x_1 \\ x_2 \end{pmatrix} - \begin{pmatrix} 0 \\ 1/2 \\ 1 \\ 0 \end{pmatrix}. \tag{30}$$

Applying equation 29 iteratively initialized with $g_0(x) = x, f_0(x) = x$ yields

$$\begin{pmatrix} g_m \\ f_m \end{pmatrix} = W_1\left(\rho\left(W_2\left(W_1\left(\rho\left(\ldots\rho\left(W_2\left(W_1\left(\rho\left(W_2\begin{pmatrix} x \\ x \end{pmatrix}\right)\right)\right)\right)\right)\right)\right)\right)\right), \tag{31}$$

and hence shows that $f_m$ can be realized through a network in $\mathcal{NN}_{m+1,4,1,1}$ with weights bounded (in magnitude) by 4. Since $\epsilon_m = 2^{-2m-2}$ and hence $\log(1/\epsilon_m) = 2m + 2$, the statement follows upon noting that $f_m(0) = 0, \forall m \in \mathbb{N}_0$. $\qquad \square$

## A.5 PROOF OF PROPOSITION 3.2

*Proof.* The proof is based on the identity

$$xy = \frac{1}{2}((x + y)^2 - x^2 - y^2), \tag{32}$$

which shows how multiplication can be implemented through the squaring operation. Let $\Psi_\delta(x)$ be a neural network approximating $x^2$ according to Proposition 3.1, i.e., $\|\Psi_\delta(x) - x^2\|_{L^\infty[0,1]} \leq \delta$,

$\Psi_\delta(0) = 0$. We first extend this approximation result to the interval $[-D, D]$, $D > 1$. Specifically, we need to find a ReLU network realization of $x^2$ and $y^2$ over $[-D, D]$ and of $(x + y)^2$ over $[-D, D]^2$. This will be accomplished by first noting that

$$\left\| 4\lceil D \rceil^2 \Psi_\delta\left(\frac{|x|}{2\lceil D \rceil}\right) - x^2 \right\|_{L^\infty([-D,D])} \leq 4\lceil D \rceil^2 \delta, \tag{33}$$

and likewise

$$\left\| 4\lceil D \rceil^2 \Psi_\delta\left(\frac{|x + y|}{2\lceil D \rceil}\right) - (x + y)^2 \right\|_{L^\infty([-D,D])^2} \leq 4\lceil D \rceil^2 \delta. \tag{34}$$

The network $x \mapsto \Psi_\delta(|x|)$ has one layer more than the network $x \mapsto \Psi_\delta(x)$ as it implements $|x| = \rho(x) + \rho(-x)$ in its first layer. Next we define for every $D \in \mathbb{R}_+$, $\delta \in (0, 1/2)$ the network

$$\Phi^*_{D,\delta}(x, y) := 2\lceil D \rceil^2 \left( \Psi_\delta\left(\frac{|x + y|}{2\lceil D \rceil}\right) - \Psi_\delta\left(\frac{|x|}{2\lceil D \rceil}\right) - \Psi_\delta\left(\frac{|y|}{2\lceil D \rceil}\right) \right), \tag{35}$$

and observe that Lemma 2.4 implies that there exists a constant $C > 0$ such that for all $D \in \mathbb{R}_+$, $\delta \in (0, 1/2)$, $x \in \mathbb{R}$ it holds that $\mathcal{L}(\Phi^*_{D,\delta}) \leq C \log(\delta^{-1})$, $\mathcal{W}(\Phi^*_{D,\delta}) = 12$, $\mathcal{B}(\Phi^*_{D,\delta}) \leq \max\{4, 2\lceil D \rceil^2\}$ and $\Phi^*_{D,\delta} x, 0) = \Phi^*_{D,\delta}(0, x) = 0$. Using Equation 32 in combination with Equation 33 and Equation 34 yields

$$\left\| \Phi^*_{D,\delta}(x, y) - \frac{1}{2}\left( (x + y)^2 - x^2 - y^2 \right) \right\|_{L^\infty([-D,D])^2} \leq 6\lceil D \rceil^2 \delta.$$

The proof is completed by taking for every $D \in \mathbb{R}_+$, $\epsilon \in (0, 1/2)$ the network $\Phi_{D,\epsilon} := \Phi^*_{D,\delta_{D,\epsilon}}$ with $\delta_{D,\epsilon} := \frac{\epsilon}{6\lceil D \rceil^2}$. □

## A.6 Proof of Lemma 3.5

*Proof.* We first consider the case $D = 1$. A fundamental result on Chebyshev interpolation, see e.g. (Liang & Srikant, 2017, Lemma 3), guarantees, for all $f \in \mathcal{S}_1$, $n \in \mathbb{N}$, the existence of a polynomial $P_{f,n}$ of degree $n$ such that

$$\|f - P_{f,n}\|_{L^\infty([-1,1])} \leq \frac{1}{2^n(n+1)!}\|f^{(n+1)}\|_{L^\infty([-1,1])} \leq \frac{1}{2^n}. \tag{36}$$

Writing the polynomials $P_{f,n}$ as $P_{f,n} = \sum_{j=0}^n a_{f,n,j}x^j$, crude—but sufficient for our purposes—estimates show that there exists a constant $c > 0$ such that for all $f \in \mathcal{S}_1$, $n \in \mathbb{N}$ it holds that

$$A_{f,n} := \max_{j=0,\dots,n} |a_{f,n,j}| \leq 2^{cn}.$$

Application of Proposition 3.3 to $P_{f,n}$ establishes the existence of a constant $C_1 > 0$ such that for all $f \in S_1$, $n \in \mathbb{N}$, $\epsilon \in (0, 1/2)$, there is a network $\Phi_{P_{f,n},1,\epsilon/2} \in \mathcal{NN}_{\infty,16,1,1}$ satisfying $\mathcal{B}(\Phi_{P_{f,n},1,\epsilon/2}) \leq \max\{A_{f,n}, 8\} \leq \max\{2^{cn}, 8\}$,

$$\mathcal{L}(\Phi_{P_{f,n},1,\epsilon/2}) \leq C_1 n(cn + \log(2/\epsilon) + \log(n)), \tag{37}$$

and

$$\|\Phi_{P_{f,n},1,\epsilon/2} - P_{f,n}\|_{L^\infty([-1,1])} \leq \frac{\epsilon}{2}. \tag{38}$$

In the following, we set $n_\epsilon = \lceil \log(2/\epsilon) \rceil$ and $\Psi_{f,\epsilon} = \Phi_{P_{f,n_\epsilon},1,\epsilon/2}$. Combining equation 36 and equation 38 establishes that for all $f \in \mathcal{S}_1$, $\epsilon \in (0, 1/2)$,

$$\|\Psi_{f,\epsilon} - f\|_{L^\infty([-1,1])} \leq \|\Psi_{f,\epsilon} - P_{f,n_\epsilon}\|_{L^\infty([-1,1])} + \|P_{f,n_\epsilon} - f\|_{L^\infty([-1,1])}$$
$$\leq \frac{\epsilon}{2} + \frac{1}{2^{n_\epsilon}} \leq \frac{\epsilon}{2} + \frac{\epsilon}{2} = \epsilon.$$

Using $\lceil \log(2/\epsilon) \rceil \leq 2\log(2/\epsilon)$ and $\log(2/\epsilon) \leq 2\log(1/\epsilon)$, for all $\epsilon \in (0, 1/2)$, in equation 37 implies the existence of a constant $C_2$ such that for all $f \in \mathcal{S}_1$, $\epsilon \in (0, 1/2)$,

$$\mathcal{L}(\Psi_{f,\epsilon}) = \mathcal{L}(\Phi_{P_{f,n_\epsilon},1,\epsilon/2}) \leq C_2(\log(\epsilon^{-1}))^2. \tag{39}$$

By the same token there exists a polynomial $\pi_1$ such that

$$\mathcal{B}(\Psi_{f,\epsilon}) = \mathcal{B}(\Phi_{P_{f,n_\epsilon},1,\epsilon/2}) \le \max\{2^{cn_\epsilon}, 8\} \le \pi_1(\epsilon^{-1}).$$

This completes the proof for the case $D = 1$.

We next prove the statement for $D \in (0,1)$. To this end, we start by noting that for $g \in \mathcal{S}_D$, with $D \in (0,1)$, the function $f_g \colon [-1,1] \to \mathbb{R}$, $x \mapsto g(Dx)$ is in $\mathcal{S}_1$. Hence, there exists, for every $g \in \mathcal{S}_D$, $\epsilon \in (0, 1/2)$, a network $\Psi_{f_g,\epsilon} \in \mathcal{NN}_{\infty,16,1,1}$ satisfying $\sup_{x \in [-1,1]} |\Psi_{f_g,\epsilon}(x) - f_g(x)| \le \epsilon$, $\mathcal{L}(\Psi_{f_g,\epsilon}) \le C_2(\log(1/\epsilon))^2$, and $\mathcal{B}(\Psi_{f_g,\epsilon}) \le \pi_1(\epsilon^{-1})$. The claim is established by taking the network approximating $g(x)$ to be $\Psi'_{f_g,\epsilon}(x) := \Psi_{f_g,\epsilon}(\frac{x}{D})$ and noting that

$$
\begin{aligned}
\sup_{x \in [-D,D]} |\Psi'_{f_g,\epsilon}(x) - g(x)| &= \sup_{x \in [-D,D]} |\Psi_{f_g,\epsilon}(\tfrac{x}{D}) - f_g(\tfrac{x}{D})| \\
&= \sup_{x \in [-1,1]} |\Psi_{f_g,\epsilon}(x) - f_g(x)| \le \epsilon,
\end{aligned}
$$

$\mathcal{L}(\Psi'_{f_g,\epsilon}) \le C_2(\log(1/\epsilon))^2$, $\mathcal{W}(\Psi'_{f_g,\epsilon}) \le 16$, and $\mathcal{B}(\Psi'_{f_g,\epsilon}) \le (1/D)\pi_1(\epsilon^{-1})$.

It remains to prove the statement for the case $D > 1$. This will be accomplished by approximating $f$ on intervals of length 2 (or less) and stitching the resulting approximations together using a localized partition of unity. To this end consider $a, b \in \mathbb{R}$ such that $1 \le b - a \le 2$, and let $h \in C^\infty([a,b], \mathbb{R})$ with $\|h^{(n)}\|_{L^\infty([a,b])} \le n!$, for all $n \in \mathbb{N}_0$. Next, note that the function $x \mapsto h\left(\frac{b-a}{2}x + \frac{b+a}{2}\right)$ is in $\mathcal{S}_1$. Hence, there exists, for every $\epsilon \in (0, 1/2)$, a network $\Psi'_{h,\epsilon} \in \mathcal{NN}_{\infty,16,1,1}$ such that $\sup_{x \in [-1,1]} |\Psi'_{h,\epsilon}(x) - h\left(\frac{b-a}{2}x + \frac{b+a}{2}\right)| \le \epsilon$, $\mathcal{L}(\Psi'_{h,\epsilon}) \le C_2(\log(1/\epsilon))^2$, and $\mathcal{B}(\Psi'_{h,\epsilon}) \le \pi_1(\epsilon^{-1})$. The networks $\Psi_{h,\epsilon}(x) := \Psi'_{h,\epsilon}\left(\frac{2}{b-a}x - \frac{b+a}{b-a}\right)$ then satisfy

$$\sup_{x \in [a,b]} |\Psi_{h,\epsilon}(x) - h(x)| = \sup_{y \in [-1,1]} |\Psi'_{h,\epsilon}(y) - h\left(\tfrac{b-a}{2}y + \tfrac{b+a}{2}\right)| \le \epsilon, \tag{40}$$

$\mathcal{L}(\Psi_{h,\epsilon}) \le C_2(\log(1/\epsilon))^2$, $\mathcal{W}(\Psi_{h,\epsilon}) \le 16$, and $\mathcal{B}(\Psi_{h,\epsilon}) \le \max\{2, |b| + |a|\}\pi_1(\epsilon^{-1})$. Now, for $D > 1$, let $N_D \in \mathbb{N}$ be such that $1 \le \frac{2D}{N_D} \le 2$ and consider the intervals

$$I_{D,k} := \left[\frac{(k-1)D}{N_D}, \frac{(k+1)D}{N_D}\right], \quad k \in \{-N_D, \ldots, N_D\}.$$

By equation 40 it follows that, for all $D > 1$, $f \in \mathcal{S}_D$, $k \in \{-N_D, \ldots, N_D\}$, and $\epsilon \in (0, 1/2)$, there exists a network $\Psi_{f,k,\epsilon} \in \mathcal{NN}_{\infty,16,1,1}$ satisfying

$$\sup_{x \in I_{D,k}} |\Psi_{f,k,\epsilon}(x) - f(x)| \le \tfrac{\epsilon}{4}, \tag{41}$$

$\mathcal{L}(\Psi_{f,k,\epsilon}) \le C_2(\log(4/\epsilon))^2$, and $\mathcal{B}(\Psi_{f,k,\epsilon}) \le \max\{2, 2|k|\}\pi_1(\epsilon^{-1})$. We next build a partition of unity through ReLU networks. Specifically, let $\chi(x) = \rho(x + 1) - 2\rho(x) + \rho(x - 1)$, set $\chi_{D,k}(x) = \chi(\frac{N_D}{D}x - k)$, $D > 1$, $k \in \mathbb{Z}$, and note that $\chi_{D,k} \in \mathcal{NN}_{2,3,1,1}$. This yields a partition of unity according to

$$\sum_{k \in \mathbb{Z}} \chi_{D,k}(x) = 1, \quad \text{for all } x \in \mathbb{R}. \tag{42}$$

For $D > 1$, $f \in \mathcal{S}_D$, $\epsilon \in (0, 1/2)$, let $f_\epsilon \colon \mathbb{R} \to \mathbb{R}$ be given by

$$f_\epsilon(x) := \sum_{k=-N_D}^{N_D} \Phi_{2,\epsilon/4}(\chi_{D,k}(x), \Psi_{f,k,\epsilon}(x)), \tag{43}$$

where $\Phi_{2,\epsilon/4}$ is the multiplication network from Proposition 3.2. Note that $|f(x)| \le 1$, for all $x \in [-D, D]$, and $|\chi_{D,k}(x)| \le 1$, for all $x \in [-D, D]$, $k \in \{-N_D, \ldots, N_D\}$. Observe further that, for each $x \in [-D, D]$, there are no more than 2 indices $k$ such that $\chi_{D,k}(x) \ne 0$. Proposition 3.2 therefore implies that the sum in equation 43 has no more than 2 non-zero terms for each $x \in [-D, D]$. Combining equation 41, equation 42, and Proposition 3.2, and noting that $\operatorname{supp}(\chi_{D,k}) = I_{D,k}$, hence yields

$$\|f_\epsilon - f\|_{L^\infty([-D,D])} \le \epsilon,$$

for all $D > 1$, $f \in \mathcal{S}_D$, $\epsilon \in (0, 1/2)$. It remains to be shown that the functions $f_\epsilon$ can be realized by networks with the desired properties. To this end, consider for every $D > 1$, $f \in \mathcal{S}_D$, $k \in \{1, \ldots, 2N_D + 1\}$, $\epsilon \in (0, 1/2)$, the network $\alpha_{f,k,\epsilon} \in \mathcal{NN}_{\infty, 19, 1, 1}$ given by

$$\alpha_{f,k,\epsilon}(x) := \Phi_{2,\epsilon/4}(\chi_{D,k-(N_D+1)}(x), \Psi_{f,k-(N_D+1),\epsilon}(x)),$$

and the network $\beta_{f,k,\epsilon} \in \mathcal{NN}_{\infty, 23, 3, 3}$ according to

$$\beta_{f,k,\epsilon}(x_1, x_2, x_3) := \begin{pmatrix} x_1 \\ \alpha_{f,k,\epsilon}(x_2) \\ x_3 \end{pmatrix}.$$

Further, set $\beta_0(x) := (x, 0, 0)^T$ and let $A \in \mathbb{R}^{3 \times 3}$ be such that $A(y_1, y_2, y_3)^T = (y_1, y_1, y_2 + y_3)^T$, for all $y_1, y_2, y_3 \in \mathbb{R}$. We can now define, for every $D > 1$, $f \in \mathcal{S}_D$, $\epsilon \in (0, 1/2)$, the network $\Psi_{f,\epsilon} \in \mathcal{NN}_{\infty, 23, 1, 1}$ given by

$$\Psi_{f,\epsilon}(x) := (0 \ 1 \ 1) \beta_{f, 2N_D+1, \epsilon}(A\beta_{f, 2N_D, \epsilon}(\ldots(A\beta_{f, 1, \epsilon}(A\beta_0(x))))).$$

Direct calculation shows that $f_\epsilon(x) = \Psi_{f,\epsilon}(x)$, for all $D > 1$, $f \in \mathcal{S}_D$, $\epsilon \in (0, 1/2)$, $x \in \mathbb{R}$. Furthermore, thanks to Proposition 3.2, there exists a constant $C_3 > 0$ such that, for all $D > 1$, $f \in \mathcal{S}_D$, $\epsilon \in (0, 1/2)$,

$$\mathcal{W}(\Psi_{f,\epsilon}) \le 4 + \max_{k \in \{1, \ldots, 2N_D+1\}} \mathcal{W}(\alpha_{f,k,\epsilon}) \le 23,$$

$$\mathcal{L}(\Psi_{f,\epsilon}) = 2 + \sum_{k=1}^{2N_D+1} \mathcal{L}(\beta_{f,k,\epsilon}) = 2 + \sum_{k=1}^{2N_D+1} \left( \mathcal{L}(\Phi_{2,\epsilon/4}) + \max\{\mathcal{L}(\chi_{k-(N_D+1)}), \mathcal{L}(\Psi_{f,k-(N_D+1),\epsilon})\} \right)$$

$$\le 2 + (2N_D + 1)(C_1 \log(16\epsilon^{-1}) + \max\{2, C_2(\log(4\epsilon^{-1}))^2\}) \le C_3 D(\log(\epsilon^{-1}))^2,$$

and

$$\mathcal{B}(\Psi_{f,\epsilon}) = \max_{k \in \{1, \ldots, 2N_D+1\}} \mathcal{B}(\alpha_{f,k,\epsilon}) \le \max\{8, 4D, \max\{2, 8D\}\pi_1(\epsilon^{-1})\} \le C'\lceil D \rceil \pi_1(\epsilon^{-1}).$$

(44)

This completes the proof. $\qquad\square$

### A.7 PROOF OF COROLLARY 4.2

*Proof.* For every $a, D \in \mathbb{R}_+$, $b \in \mathbb{R}$, $\epsilon \in (0, 1/2)$ take the network given by $\Psi_{a,b,D,\epsilon}(x) := \Psi_{a, D + \frac{|b|}{a}, \epsilon}(x - \frac{b}{a})$ with $\Psi_{a, D + \frac{|b|}{a}, \epsilon}$ according to Theorem 4.1 and observe that

$$\sup_{x \in [-D, D]} |\Psi_{a,b,D,\epsilon}(x) - \cos(ax - b)| = \sup_{[-(D + \frac{|b|}{a}), D + \frac{|b|}{a}]} |\Psi_{a, D + \frac{|b|}{a}, \epsilon}(y) - \cos(ay)| \le \epsilon.$$

Applying Theorem 4.1 completes the proof.

$\qquad\square$

### A.8 PROOF OF PROPOSITION 5.2

*Proof.* For all $D, a \in \mathbb{R}_+$, $f \in \mathcal{F}_{D,a}$, let $g_f, h_f \in \mathcal{S}_D$ be functions such that $f = \cos(ag_f)h_f$ holds. Note that Lemma 3.5 guarantees the existence of a constant $C_1 > 0$ and a polynomial $\pi_1$ such that for all $D, a \in \mathbb{R}_+$, $f \in \mathcal{F}_{D,a}$, $\epsilon \in (0, 1/2)$ there are networks $\Psi_{h_f,\epsilon}, \Psi_{g_f,\epsilon} \in \mathcal{NN}_{\infty, 23, 1, 1}$ which satisfy $\mathcal{L}(\Psi_{h_f,\epsilon}), \mathcal{L}(\Psi_{g_f,\epsilon}) \le C_1 \lceil D \rceil (\log([\frac{\epsilon}{12\lceil a \rceil}]^{-1}))^2$, $\mathcal{B}(\Psi_{h_f,\epsilon}), \mathcal{B}(\Psi_{g_f,\epsilon}) \le \max\{1/D, \lceil D \rceil \pi_1([\frac{\epsilon}{12\lceil a \rceil}]^{-1})\}$ and

$$\|\Psi_{g_f,\epsilon} - g_f\|_{L^\infty([-D,D])}, \quad \|\Psi_{h_f,\epsilon} - h_f\|_{L^\infty([-D,D])} \le \frac{\epsilon}{12\lceil a \rceil}. \tag{45}$$

Theorem 4.1 further ensures the existence of a constant $C_2 > 0$ such that for all $a, D \in \mathbb{R}_+$, $\epsilon \in (0, 1/2)$ there is a neural network $\Phi_{a,D,\epsilon} \in \mathcal{NN}_{\infty, 16, 1, 1}$ which satisfies $\mathcal{L}(\Phi_{a,D,\epsilon}) \le C_2((\log(1/\epsilon))^2 + \log(\lceil aD \rceil))$, $\mathcal{B}(\Phi_{a,D,\epsilon}) \le C_2$, and

$$\|\Phi_{a,D,\epsilon} - \cos(a \cdot)\|_{L^\infty([-D,D])} \le \frac{\epsilon}{3}. \tag{46}$$

Further, thanks to Proposition 3.2, there exists a constant $C_3 > 0$ such that for all $\epsilon \in (0, 1/2)$ there is a network $\mu_\epsilon \in \mathcal{NN}_{\infty, 12, 2, 1}$ which satisfies $\mathcal{L}(\mu_\epsilon) \leq C_3 \log(1/\epsilon)$, $\mathcal{B}(\mu_\epsilon) \leq \max\{4, 2\lceil D \rceil^2\}$, and

$$\sup_{x, y \in [-D, D]} |\mu_\epsilon(x, y) - xy| \leq \tfrac{\epsilon}{3}. \tag{47}$$

For all $D, a \in \mathbb{R}_+$, $f \in \mathcal{F}_{D, a}$, $\epsilon \in (0, 1/2)$ we define the neural networks

$$\Gamma_{f, \epsilon} := \mu_\epsilon(\Phi_{a, D, \epsilon}(\Psi_{g_f, \epsilon}), \Psi_{h_f, \epsilon}). \tag{48}$$

First, observe that Equation 45, Equation 46, and $\sup_{x \in \mathbb{R}} |\frac{d}{dx} \cos(ax)| = a$ imply for all $x \in [-D, D]$ that

$$\begin{aligned}
|\Phi_{a, D, \epsilon}(\Psi_{g_f, \epsilon}(x)) - \cos(a g_f(x))| &\leq |\Phi_{a, D, \epsilon}(\Psi_{g_f, \epsilon}(x)) - \cos(a \Psi_{g_f, \epsilon}(x))| \\
&\quad + |\cos(a \Psi_{g_f, \epsilon}(x)) - \cos(a g_f(x))| \\
&\leq \tfrac{\epsilon}{3} + a \tfrac{\epsilon}{12\lceil a \rceil} \leq \tfrac{5\epsilon}{12}.
\end{aligned}$$

Combining this with Equation 45, Equation 47, and $\|\cos\|_{L^\infty[-D, D]}, \|f\|_{L^\infty[-D, D]} \leq 1$ yields for all $x \in [-D, D]$ that

$$\begin{aligned}
|\Gamma_{f, \epsilon}(x) - f(x)| &= |\mu_\epsilon(\Phi_{a, D, \epsilon}(\Psi_{g_f, \epsilon}(x)), \Psi_{h_f, \epsilon}(x)) - \cos(a g_f(x)) h_f(x)| \\
&\leq |\mu_\epsilon(\Phi_{a, D, \epsilon}(\Psi_{g_f, \epsilon}(x)), \Psi_{h_f, \epsilon}(x)) - \Phi_{a, D, \epsilon}(\Psi_{g_f, \epsilon}(x)) \Psi_{h_f, \epsilon}(x)| \\
&\quad + |\Phi_{a, D, \epsilon}(\Psi_{g_f, \epsilon}(x)) \Psi_{h_f, \epsilon}(x) - \cos(a g_f(x)) h_f(x)| \\
&\leq \tfrac{\epsilon}{3} + \tfrac{5\epsilon}{12} + \tfrac{\epsilon}{12\lceil a \rceil} + \tfrac{5\epsilon}{12} \tfrac{\epsilon}{12\lceil a \rceil} \leq \epsilon.
\end{aligned}$$

By construction there exists a constant $C_4$ and a polynomial $\pi_2$ such that for all $D, a \in \mathbb{R}_+$, $f \in \mathcal{F}_{D, a}$, $\epsilon \in (0, 1/2)$ it holds that $\mathcal{W}(\Gamma_{f, \epsilon}) = 46$,

$$\mathcal{L}(\Gamma_{f, \epsilon}) \leq \mathcal{L}(\mu_\epsilon) + \max\{\mathcal{L}(\Phi_{a, D, \epsilon}) + \mathcal{L}(\Psi_{g_f, \epsilon}), \mathcal{L}(\Psi_{h_f, \epsilon})\} \leq C_4 \lceil D \rceil ((\log(\epsilon^{-1}))^2 + \log(\lceil a \rceil)),$$

and

$$\mathcal{B}(\Gamma_{f, \epsilon}) \leq \max\{\mathcal{B}(\mu_\epsilon), \mathcal{B}(\Phi_{a, D, \epsilon}), \mathcal{B}(\Psi_{g_f, \epsilon}), \mathcal{B}(\Psi_{h_f, \epsilon})\} \leq \max\{1/D, \pi_2(\epsilon^{-1}, \lceil D \rceil, \lceil a \rceil)\}.$$

This completes the proof. $\qquad \square$

### A.9 PROOF OF PROPOSITION 5.3

*Proof.* For every $N \in \mathbb{N}$, $p \in (0, 1/2)$, $a \in \mathbb{R}_+$, $x \in \mathbb{R}$, let $S_{N, p, a}(x) = \sum_{k=0}^{N} p^k \cos(a^k \pi x)$. The geometric sum formula ensures that

$$|S_{N, p, a}(x) - W_{p, a}(x)| \leq \sum_{k=N+1}^{\infty} |p^k \cos(a^k \pi x)| \leq \sum_{k=N+1}^{\infty} p^k = \tfrac{1}{1-p} - \tfrac{1-p^{N+1}}{1-p} \leq 2^{-N}. \tag{49}$$

Let $N_\epsilon := \lceil \log(2/\epsilon) \rceil$, $\epsilon \in (0, 1/2)$. Next note that Theorem 4.1 ensures the existence of a constant $C_1 > 0$ such that for all $a, D \in \mathbb{R}_+$, $k \in \mathbb{N}_0$, $\epsilon \in (0, 1/2)$ there is a network $\phi_{a^k, D, \epsilon} \in \mathcal{NN}_{\infty, 16, 1, 1}$ which satisfies $\mathcal{L}(\phi_{a^k, D, \epsilon}) \leq C_1((\log(\epsilon^{-1}))^2 + \log(\lceil a^k \pi D \rceil))$, $\mathcal{B}(\phi_{a^k, D, \epsilon}) \leq C_1$, and

$$\|\phi_{a^k, D, \epsilon} - \cos(a^k \pi \cdot)\|_{L^\infty([-D, D])} \leq \tfrac{\epsilon}{4}. \tag{50}$$

Thanks to $x = \rho(x) - \rho(-x)$ and Lemma 2.3, there exists, for every $L \in \mathbb{N}$, a neural network $\tau_L \in \mathcal{NN}_{L, 2, 1, 1}$ which satisfies for all $x \in \mathbb{R}$ that $\tau_L(x) = x$. For all $p \in (0, 1/2)$, $a, D \in \mathbb{R}_+$, $k \in \mathbb{N}_0$, $\epsilon \in (0, 1/2)$, define the neural networks

$$\psi_{D, \epsilon}^{p, a, 0}(x) = \begin{pmatrix} x \\ p^0 \phi_{a^0, D, \epsilon}(x) \\ 0 \end{pmatrix} \quad \text{and} \quad \psi_{D, \epsilon}^{p, a, k}(x_1, x_2, x_3) = \begin{pmatrix} x_1 \\ p^k \phi_{a^k, D, \epsilon}(x_2) \\ x_3 \end{pmatrix}, \, k > 0, \tag{51}$$

and let $A \in \mathbb{R}^{3 \times 3}$ be such that $A(y_1, y_2, y_3)^T = (y_1, y_1, y_2 + y_3)^T$, for all $y \in \mathbb{R}^3$. Consider now, for $p \in (0, 1/2)$, $a, D \in \mathbb{R}_+$, $\epsilon \in (0, 1/2)$, the network $\Psi_{p, a, D, \epsilon}$ defined by

$$\Psi_{p, a, D, \epsilon}(x) := (0 \quad 1 \quad 1) \psi_{D, \epsilon}^{p, a, N_\epsilon}(A \psi_{D, \epsilon}^{p, a, N_\epsilon - 1}(\dots (A \psi_{D, \epsilon}^{p, a, 0}(x)))). \tag{52}$$

Note Equation 50 combined with the geometric sum formula implies that for all $p \in (0, 1/2)$, $a, D \in \mathbb{R}_+$, $\epsilon \in (0, 1/2)$, $x \in [-D, D]$ we have

$$|\Psi_{p,a,D,\epsilon}(x) - S_{N_\epsilon,p,a}(x)| = \left| \sum_{k=0}^{N_\epsilon} p^k \phi_{a^k,D,\epsilon}(x) - \sum_{k=0}^{N_\epsilon} p^k \cos(a^k \pi x) \right|$$

$$\leq \sum_{k=0}^{N_\epsilon} p^k |\phi_{a^k,D,\epsilon}(x) - \cos(a^k \pi x)| \leq \tfrac{\epsilon}{4} \sum_{k=1}^{\infty} 2^{-k} = \tfrac{\epsilon}{2}.$$

Combining this with Equation 49 establishes that for all $p \in (0, 1/2)$, $a, D \in \mathbb{R}_+$, $\epsilon \in (0, 1/2)$, $x \in [-D, D]$ it holds

$$|\Psi_{p,a,D,\epsilon}(x) - W_{p,a}(x)| \leq 2^{-\lceil \log(\frac{2}{\epsilon}) \rceil} + \tfrac{\epsilon}{2} \leq \tfrac{\epsilon}{2} + \tfrac{\epsilon}{2} = \epsilon.$$

By construction there exists a constant $C_2$ such that for all $p \in (0, 1/2)$, $a, D \in \mathbb{R}_+$, $\epsilon \in (0, 1/2)$ we have $\mathcal{W}(\Psi_{p,a,D,\epsilon}) = 20$,

$$\mathcal{L}(\Psi_{p,a,D,\epsilon}) \leq \sum_{k=0}^{N_\epsilon} \mathcal{L}(\phi_{a^k,D,\epsilon}) \leq (N_\epsilon + 1) C_1 ((\log(\epsilon^{-1}))^2 + \log(\lceil a^{N_\epsilon} \pi D \rceil))$$

$$\leq C_2 ((\log(\epsilon^{-1}))^3 + (\log(\epsilon^{-1}))^2 \log(\lceil a \rceil) + \log(\epsilon^{-1}) \log(D)),$$

and

$$\mathcal{B}(\Psi_{p,a,D,\epsilon}) = \max_{k \in \{0,\dots,N_\epsilon\}} \mathcal{B}(\phi_{a^k,D,\epsilon}) = C_1.$$

This completes the proof.

$\square$

### A.10   PROOF OF PROPOSITION 6.4

*Proof.* First note that for every polynomial $\tilde{\pi}$ it holds that $\tilde{\pi}(\log(t)) \in \mathcal{O}(t)$, $t \to \infty$. Since $x \mapsto (2\pi(x))^L$ is a polynomial, there exists $a \in \mathbb{N}$ such that $a > (2\pi(\log(a)))^L$. Lemma 6.2 now implies that any network $\Phi \in \mathcal{NN}_{L,M,1,1}$ with $M \leq \pi(\log(a))$ is $(2\pi(\log(a)))^L$-sawtooth and therefore has less than $a$-many different linear pieces. Hence there exists an interval $[u_1, u_2] \subseteq [0, u]$ with $u_2 - u_1 \geq (u/a)$ on which $\Phi$ is linear. Since $u_2 - u_1 \geq (u/a)$ the interval supports a full period of $f(a \cdot)$ and we can therefore conclude that

$$\|f(a \cdot) - \Phi\|_{L^\infty([0,u])} \geq \|f(a \cdot) - \Phi\|_{L^\infty([u_1,u_2])} \geq \inf_{\substack{\delta \in [0,u), \\ c,d \in \mathbb{R}}} \|f(x) - (cx + d)\|_{L^\infty([\delta, u+\delta])} = \xi(f).$$

Finally note, that $\xi(f) > 0$ holds by assumption, since any continuous $u$-periodic function which is linear on an interval of length $u$ must be constant.

$\square$

### A.11   PROOF OF THEOREM 6.6

*Proof.* The proof will be effected by contradiction. Assume that for every $\epsilon > 0$ there exists a network $\Phi_\epsilon \in \mathcal{NN}_{L,M,1,1}$ with $M \leq \pi(\log(\epsilon^{-1}))$ and $\|f - \Phi_\epsilon\|_{L^\infty([a,b])} \leq \epsilon$. Since every ReLU network is piecewise linear we can now apply Theorem 6.5 to conclude that there exists a constant $C$ such that for all $\epsilon > 0$ the network $\Phi_\epsilon$ must have at least $C\epsilon^{-\frac{1}{2}}$ many different linear pieces. This leads to a contradiction as, by assumption combined with Lemma 6.2, $\Phi_\epsilon$ is $(2\pi(\log(\epsilon^{-1})))^L$-sawtooth and it holds for every polynomial $\tilde{\pi}$ that $\tilde{\pi}(\log(\epsilon^{-1})) \in o(\epsilon^{-1/2})$, $\epsilon \to 0$. $\square$

