# OpenReview forum: "The Universal Approximation Power of Finite-Width Deep ReLU Networks"
_ICLR.cc/2019/Conference_

### Official Review · AnonReviewer2 · 2018-11-02
**narrow choice of function space, unclear how it relates to holder, sobolev**

**Rating:** 6
**Confidence:** 3

**Review:**

The focus of this paper is to show that finite-width deep neural networks with fully connected layers and ReLU activations are rate-distortion optimal approximators of certain classes of functions, meaning the approximation error decays exponentially in the number of neurons in the network. The function classes explored in this paper are: 1-d polynomials (on bounded intervals), 1-d sinusoidal functions (on bounded intervals), and other 1-d functions built from compositions or linear combinations of these, such as the so-called class of “oscillatory textures” and a class of continuous but nowhere differentiable functions known as Weierstrass functions. Finally, the paper also shows that as the desired approximation accuracy goes to zero finite-width deep ReLU networks require asymptotically fewer neurons than finite-depth wide ReLU networks in approximating a broad class of smooth functions.


The paper is well-written and the technical results are presented in a way that is easy to understand. The results are somewhat novel, although they do build off other recent works, namely Yarotsky (2016) and Telgarsky (2015). However, the authors were careful to cite when they reuse proof techniques from these and other works. The results in the main text appear to be technically sound. I did not check carefully all the proofs in the supplemental materials.


My major criticism is that the focus on certain specific function classes (oscillatory textures, Weierstrauss functions) seems arbitrary, and leaves open many questions. For example, there is existing work on the approximation ability of deep ReLU networks for functions in more general Holder and Sobolev spaces:

Hadrien Montanelli and Qiang Du. Deep ReLU networks lessen the curse of dimensionality. arXiv preprint arXiv:1712.08688, 2017.

J. Schmidt-Hieber. Nonparametric regression using deep neural networks with ReLU activation function. ArXiv e-prints, August 2017.

I was left wondering how the present results relate to these works, and what insight we get from understanding these particular function classes that we don't get from understanding Holder or Sobolev spaces.


Major comments


In Section 3, I found the progression of the results from approximation of x^2, to multiplication xy, and to general smooth functions to be very natural and well-motivated. However, sections 4 and 5 seem lack somewhat in motivation, since here the authors focus on very specific function classes (sinusoidal functions, oscillatory textures, and Weierstrass functions). While these results are still interesting, focusing on such specific functions is less satisfactory, since it raises questions about the true scope of the results (e.g., will similar approximation rates extend to other fractal functions, or just Weierstrauss functions?). Could the authors give further justification for why these function classes are interesting to focus on, or why they limit themselves in this way? Can the authors also put these results more into context with existing results on the approximation with ReLU networks?


The authors state multiple times that “all our results apply to the multivariate case” but that they restrict themselves to the univariate case for simplicity of presentation. While this is fine, some indication of how the results are altered in the multivariate case would be useful. For example, does the fixed-width M in multivariate generalizations of Prop 3.1--3.3 need to be bigger, smaller, or the same? What other constants are dimensionally dependent? Do the multivariate generalization of their results bear the "curse of dimensionality", i.e., does the number of neurons needed to reach epsilon accuracy depend geometrically on the dimension?

Minor comments


A conclusion or discussion section summarizing the overall technical contribution would be useful for the reader. Also, it would be useful to include some discussion on remaining open problems or future work.

On pg. 2, the authors state “the approximation results throughout the paper guarantee that the magnitude of the weights in the network does not grow faster than polynomially in the cardinality of of the domain over which the approximation takes place”. What does “cardinality of the domain” here mean? I think the authors mean the size D of the interval [-D,D] over which the approximation is valid.

On pg. 7, the authors say “We note that this result allows to show that local cosine bases (cite) can be approximated by deep ReLU networks with exponential error decay…”. I think the authors mean to say “...this result allows us to show…” or “this result allows one to show…”. Although it’s not clear to me whether this means it has been shown (it’s a direct corollary), or could possible be shown (it’s a corollary, but needs some non-trivial work). Also, one line to specify what a “local cosine basis” is would be helpful.

---

> ### Author Response · Authors · 2018-11-26
> **No known classical methods that would yield exponential approximation accuracy for Weierstrass function and oscillatory textures**
>
> Thank you for pointing out references (Montanelli et al., 2017) and (Schmidt-Hieber, 2017).
>
> We considered the Weierstrass function and oscillatory textures as these functions are known to be hard to approximate and there are no known classical methods that would yield exponential approximation accuracy. The main point of our results here is that deep ReLU networks do provide exponential approximation accuracy and are hence the first approximation method to deliver exponential accuracy. Moreover, the Weierstrass function does not have weak derivatives and possesses Hölder smoothness which can be infinitely small; it can therefore not be considered an element of a particular Hölder or Sobolev space. Regarding our other results, we achieve faster approximation rates than the ones obtained for general Sobolev and Hölder spaces.
>
> The ReLU network approximation results for functions in the unit ball in Sobolev spaces first obtained in (Yarotsky, 2017) and cited in (Montanelli et al., 2017) show that the number of nonzero weights needed in the approximating ReLU network grows “polynomially” in the inverse of the approximation error, e.g. O(\eps^{-d/m} \log(1/eps)) for W^{m,\infty}([0,1]^d).
>
> Theorem 5 in (Schmidt-Hieber, 2017) also states that in the approximation of functions in the unit ball in Hölder spaces, the number of nonzero weights needed in the approximating ReLU network has to grow “polynomially” in the inverse of the approximation error.
>
> In contrast, we show
>
> 1. Exponential approximation accuracy for highly oscillatory sinusoidal functions (Theorem 4.1), leading to exponential approximation accuracy for oscillatory textures (Proposition 5.2) and the Weierstrass function (Proposition 5.4). Exponential approximation accuracy means that the approximation error decays exponentially in the number of nonzero weights, which is equivalent to the number of nonzero weights required growing “logarithmically” in the inverse approximation error.
> 2. Our results are stated for arbitrary domains, in contrast to a restriction to the unit ball in (Yarotsky, 2017), (Montanelli et al., 2017) and (Schmidt-Hieber, 2017).
> 3. We establish that the weights in the networks we construct grow no faster than polynomial in the size of the domain over which approximation is carried out, a result that is crucial to be able to conclude exponential approximation accuracy. Such a result is not available in existing work (Yarotsky, 2017), (Montanelli et al., 2017) and (Schmidt-Hieber, 2017).
>
> These three novel aspects allow us to then conclude that the approximation results we found are best possible in the sense of (Bölcskei et al., 2017).
>
>
> --Major comments--
>
> On the motivation to focus on specific function classes
>
> We find these results interesting as they demonstrate that deep ReLU networks can deliver something that to date has not been accomplished, namely to approximate (certain) fractal functions and oscillatory textures with exponential accuracy. We chose these two function classes as they are known to be notoriously hard to approximate. In fact, the best approximation results available in the literature for these function classes exhibit low-order polynomial approximation accuracy, i.e., polynomial error decay as opposed to exponential error decay.
>
> In practice, the functional dependence a network is to learn is unknown. In this sense our results say that even very unusual functional dependencies can be learned, in fact, optimally so in an approximation-theoretic sense, by a deep ReLU network, thus speaking to the universal approximation capability of deep ReLU networks.
>
> The results on the approximation of the Weierstrass function can be extended in a straightforward way to certain fractal functions of the same nature, e.g. the Blancmange curve. The generalization to all fractal functions remains an open problem.
>
>
> On the curse of dimensionality
>
> The multivariate generalizations of Props. 3.1 & 3.2 follow straightforwardly from the univariate case. For the multivariate generalization of Prop. 3.3 one can show that the width grows linearly, and hence optimally, in the number of dimensions.
>
> The multivariate generalizations of our results do not suffer from a “curse of dimensionality”; in fact, our results exhibit the same qualitative behavior as those reported previously in Prop. 3.3 and Lemma 3.8 in (Schwab&Zech, 2017), which show that the “curse of dimensionality” can be overcome when approximating polynomials by ReLU networks.
>
> In addition, it was shown recently in (Grohs et al., 2018)  that deep ReLU networks can approximate certain solution families of parametric PDEs depending on a large (possibly infinite) number of parameters while overcoming the curse of dimensionality.

---

> ### Author Response · Authors · 2018-11-26
> **Minor comments**
>
> --Minor comments--
>
> On future work
>
> Thank you for the suggestion, we will include such a discussion.
>
>
> On "cardinality of the domain"
>
> Yes, you are right. Thank you for pointing this out. We corrected it in the updated version of the paper.
>
>
> On local cosine bases
>
> In order to establish the approximability of local cosine bases by deep ReLU networks, we need three components---approximability of sinusoidal functions, approximability of the window function, and approximability of their product. While approximability of sinusoidal functions and the product function is shown in our work (Theorem 4.1 and Proposition 3.2), the approximability of window functions needs to be established separately. We have shown approximability for Wilson bases with a compactly supported window function and for Gabor systems with a Gaussian window function. These results will be reported elsewhere.
>
> The result in Proposition 5.2 is restricted to window functions in S_D (Def. 3.4) and does not apply to arbitrary functions. As we have a more general result, though not reported in this paper, we decided to remove this comment in the revised version.

---

### Official Review · AnonReviewer3 · 2018-11-04
**approximation theory using relu networks**

**Rating:** 5
**Confidence:** 4

**Review:**

This paper describes results regarding approximations of certain function families using ReLU neural networks. The authors emphasize two points about these networks: finite width and depth that is logarithmic in the approximation error parameter $\epsilon$.

The first result concerns approximation of polynomials, which is used as a building block for all subsequent results. This result itself is quite simple and mostly follows from simple observations or known results, though it is possible that these have not been explicitly written in this form anywhere. The other results concern smooth functions, and some kinds of non-smooth functions such as the Weirstrass function. There are two neat observations (i) using the sawtooth function to approximate sinusoidal ones and (ii) using overlapping "approximation" to simulate an indicator.

The paper is refreshingly well-written and pleasant to read. Most of the results are tailored to work for either periodic functions, or can be expressed as: if piecewise polynomials are a good approximation, then so are constant depth neural networks with ReLU. I'm not sure that ICLR is the best venue for these kinds of results, as any connection with learning is at best tenuous, and the kind of approximation results don't seem to have any direct bearing on machine learning.

---

> ### Author Response · Authors · 2018-11-26
> **We show that ReLU networks can learn unusual functional dependencies optimally in an approximation-theoretic sense**
>
> ******************************************************************************************
> The first result concerns approximation of polynomials, which is used as a building block for all subsequent results. This result itself is quite simple and mostly follows from simple observations or known results, though it is possible that these have not been explicitly written in this form anywhere. The other results concern smooth functions, and some kinds of non-smooth functions such as the Weirstrass function. There are two neat observations (i) using the sawtooth function to approximate sinusoidal ones and (ii) using overlapping "approximation" to simulate an indicator.
> ******************************************************************************************
>
>
> We see the following novel elements in our paper.
>
> Proposition 3.3:
>
> 1) We do not need skip connections, in contrast to (Yarotsky, 2016).
> 2) Our results are stated for arbitrary domains, in contrast to a restriction to the unit cube in (Yarotsky, 2016).
> 3) The approximating network we construct has fixed width 16, independently of the degree of the polynomial it approximates. In contrast, the width of the approximating network in (Yarotsky, 2016) scales with the order of the polynomial.
> 4) We establish that the weights in the approximation network we construct grow no faster than polynomial in the size of the domain over which approximation is carried out. This allows us to then conclude that our approximating network provides exponential approximation accuracy and is hence best possible in an approximation-theoretic sense.
>
> Moreover, our Proposition 3.3 enables us to show how—based on Telgarsky’s and Yarotsky’s sawtooth construction—deep ReLU networks can approximate highly oscillatory sinusoidal functions (Theorem 4.1), leading to exponential approximation rates for oscillatory textures (Proposition 5.2) and the Weierstrass function (Proposition 5.4), both of which are known to be poorly approximated by classical methods.
>
> Additionally, we consider the depth-width tradeoff analysis in Section 6 novel.
>
> To the best of our knowledge none of these results are available in existing works.
>
>
> ******************************************************************************************
> The paper is refreshingly well-written and pleasant to read. Most of the results are tailored to work for either periodic functions, or can be expressed as: if piecewise polynomials are a good approximation, then so are constant depth neural networks with ReLU. I'm not sure that ICLR is the best venue for these kinds of results, as any connection with learning is at best tenuous, and the kind of approximation results don't seem to have any direct bearing on machine learning.
> ******************************************************************************************
>
>
> As the ICLR 2019 - Call For Papers listed “theoretical issues in deep learning” as one of the conference topics, we felt that the paper could be a fit.
>
> The relation to learning, in our view, is as follows. In practice, the functional dependence a network is to learn is unknown. In this sense our results say that a broad class of functional dependencies, even very unusual ones such as oscillatory textures and certain fractal functions, can be learned, in fact, optimally so in an approximation-theoretic sense, by a deep ReLU network, thus speaking to their universal approximation capability.

---

### Official Review · AnonReviewer4 · 2018-11-19
**The paper considers uniform approximations of functions by ReLU neural networks. The authors derive bounds on the depth and width of such networks to achieve a certain degree of accuracy (\eps) over classes of functions such as polynomials, sinusoidals, oscillatory textures, etc.**

**Rating:** 5
**Confidence:** 3

**Review:**

A main theme of the paper is showing that constant-width ReLU networks with depth increasing ploy-logarithmically in 1/\eps can achieve the desired accuracy over the classes considered.

- It seems to me that a major claim of the paper is that previous results did not have constant-width approximations.

However, this does not seem accurate to me. For example, for polynomials, the statement “the width of the approximating network does not grow with the degree of the polynomial as is the case in Yarotsky (2016)...” does not seem true. The constructions in Yarotsky (2016) which much of the present work appears to be based on, in fact, allow for a constant-width approximation of polynomials of degree “n” in “d”-variables, over the cube, with

wdith = 9, and depth ~ m \log m [\log (1/\eps) + \log m] where m = n + d. (*)

This bound is quite similar to Prop. 3.3 (with maybe even sharper dependence on “m”: m(\log m)^2 versus m^2 in the paper. PS. I would also  double-check C in Prop. 3.3 which could be growing logarithmically in m.)

(*) can be seen by inspecting the arguments around (14)-(15) in Yarotsky (2016) and noting that c_1 ~ \log m in that argument. For example, considering d=1 which is the focus of the present paper, Yarotsky (2016) shows a constant-width approximation to the product function (x,y) \mapsto xy which can be used to build a constant width-approximation to the monomial of the highest degree in the polynomial by recursive composition. All other monomials can be accessed serially at various depths of that architecture.

- Much of the subsequent results in the paper are based on this constant-width approximation of polynomials as the authors point out. This is not that surprising given the Taylor approximation. For example, in Theorem 4.1, the first few lines of the proof show that the cosine can be approximated with a polynomial of degree m = O(\log (1/\eps)). Combining this with the polynomial approximation result one gets a constant-width approximation with
depth ~ \log(1/\eps)^2 or \log(1\eps)^2 \log \log (1/\eps)
depending which version of the polynomial approximation result one believes (m \log m versus m as the prefactor as discussed above).  In other words, it appears to me that the proof of Theorem 4.1 can be shortened considerably. It would be a corollary of the polynomial approximation.

- A novelty of the present work over Yarotsky (2016) that the authors point out is avoiding skip connections in Proposition 3.1. (Perhaps this is true also for Prop. 3.3? Not discussed.) I haven’t checked the details here, but assuming correctness, I agree that it is quite interesting. I am not sure however if it is a significant improvement over the existing results.

- The discussion of Section 5 and 6 might be new in this context and somewhat interesting. However, at least that of Section 5 again seems to be a natural byproduct the polynomial approximation result.

- I would like to point out that there are other results on finite-width approximation by ReLU networks, establishing some quite sharp bounds, for example,
Hanin and Sellke 2017: arXiv:1710.11278
Yarotsky 2018: arXiv:1802.03620
In light of these, it wouldn’t be that accurate to claim that the issue of constant-width approximation is considered for the first time in the present paper.

---

> ### Author Response · Authors · 2018-11-26
> **Taylor approximation is not the main idea of the proof**
>
> On the width in (Yarotsky, 2016)
>
> Note that in the arguments around (14)-(15) in Yarotsky (2016) only the term p_{m,n} = (x - m/N)^n is approximated. The entire polynomial p = \sum_m \sum_n  a_{m,n} (x - m/N)^n is approximated by a network consisting of parallel subnetworks that approximate the individual p_{m,n}, see the discussion below (18) in Yarotsky (2016). Therefore, the width of the network in Yarotsky (2016) approximating the overall polynomial grows with the number of non-zero coefficients a_{m,n} and hence does not remain constant for arbitrary polynomials, in fact, in the worst case (i.e., all a_{m,n} \neq 0) it grows with the degree.
>
> We avoid this network parallelization and the associated width growth by iteratively composing multiplication networks and combining them with a construction that computes higher order monomials together with a lower order polynomial. This construction, to the best of our knowledge, is novel.
>
> We suppose that the width of 9 as suggested by the reviewer is based on the assumption that multiplication could be implemented by parallelizing 3 networks with skip connections, each of width 3 and approximating x^2. While this might be true for networks with skip connections, it does not apply to our setting where skip-connections are not allowed.
>
>
> On the cosine approximation
>
> We would like to point out that the Taylor approximation was only a minor part of the proof. The main idea in the proof of Theorem 4.1 is a novel approach which shows how Telgarsky’s and Yarotsky’s “sawtooth construction” can be used to approximate sinusoidal functions with networks of depth scaling logarithmically in the frequency of the sinusoid. Applying the Taylor expansion straightforwardly as suggested by the reviewer would lead to network weights growing with the frequency of the sinusoid, which would, in turn, lead to a weight growth that violates the weight growth behavior we require to be able to conclude rate-distortion optimality in the sense of (Bölcskei et al., 2017).
>
>
> On the skip-connections and improving on the existing results
>
> All the results in our paper, indeed, apply to networks without skip connections, see Def. 2.1.
>
> Propositions 3.1-3.3 go beyond Yarotsky’s work as follows:
> 1. (Props. 3.1 & 3.2) The approximating networks for the squaring and multiplication operations we construct do not need skip connections.
> 2. (Prop. 3.3) Our results are stated for arbitrary domains, in contrast to a restriction to the unit ball in (Yarotsky, 2017).
> 3. (Prop. 3.3) The approximating network we construct has fixed width, independently of the degree of the polynomial it approximates. In contrast, the width of the approximating network in (Yarotsky, 2016) scales with the order of the polynomial.
> 4. (Prop.  3.2) We establish that the weights in the multiplication network we construct grow no faster than polynomial in the size of the domain over which approximation is carried out. This allows us to then conclude that our approximating network provides exponential approximation accuracy and is hence best possible in an approximation-theoretic sense (see (Bölcskei et al., 2017)).
>
>
> On the Sections 5 and 6
>
> The main idea in the proof of Theorem 4.1 is a novel approach which shows how Telgarsky’s and Yarotsky’s “sawtooth construction” can be used to approximate sinusoidal functions with networks of depth scaling logarithmically in the frequency of the sinusoid. Applying the Taylor expansion straightforwardly as suggested by the reviewer would lead to network weights growing with the frequency of the sinusoid, which would, in turn, lead to a weight growth that violates the weight growth behavior we require to be able to conclude rate-distortion optimality in the sense of (Bölcskei et al., 2017).
>
> In Section 5, we show that Theorem 4.1 leads to exponential approximation rates and rate-distortion optimality in the sense of (Bölcskei et al., 2017) for oscillatory textures (Proposition 5.2) and for the Weierstrass function (Proposition 5.4), both of which are known to be poorly approximated by classical methods. In fact, the best known approximation results exhibit polynomial approximation rates only.
>
>
> On the existing work on finite-width networks (Hanin&Sellke, 2017) and (Yarotsky, 2018)
>
> We thank the reviewer for pointing out these references, which, indeed, contain results on finite-width approximations of continuous functions. In contrast to our results, these results do, however, not say anything about the depth of the approximating networks and do not provide  bounds on their weights’ magnitudes. These aspects are important, as it is the very combination of finite width, depth scaling poly-logarithmically in 1/\eps, and weights growing no faster than polynomially in 1/\eps that guarantees rate-distortion optimality in the sense of Bölcskei et al. (2017) and consequently exponential error decay
>
> We will cite the papers mentioned by the reviewer in the revised version of the paper.

---

### Meta-Review · Area_Chair1 · 2018-12-17
**Well written paper, missing a clear selling point**

**Confidence:** 4
**Recommendation:** Reject

**Metareview:**

The paper contributes to the theoretical understanding of finite width ReLU networks. It contributes new ideas and constructions to investigate the representational power of such networks. In particular, the analysis works without skip connections. Referees found the paper refreshingly well-written and pleasant to read.

There is a concern that the paper may be overstating the novelty and innovation of the results, as some of them are easy implications, and there are other previous works that have obtained results on finite width networks (see AnonReviewer4's comments).  On the other hand, the authors were careful to cite when they reuse proof techniques from these and other works (AnonReviewer2). Another concern is that the considered target function space might be too narrow (see AnonReviewer2's comments). The authors clarify that the choice was because the considered classes are known to be hard to approximate and there are no known classical methods that would yield exponential approximation accuracy. Another concern is that the results might not be suitable to ICLR, having an emphasis on approximation theory and less on learning (see AnonReviewer3's comments).

The reviewers consistently rate the paper as not very strong, with one marginally above acceptance threshold and two marginally below acceptance threshold ratings.

While this appears to be a well written paper with valuable new ideas in regard to the approximation properties of networks, the contributions were not convincing enough. I would suggest that developing a clearer connecting to learning and broader classes of target functions could increase the appeal of the paper.